# Microbiome depletion and recovery in the sea anemone, *Exaiptasia diaphana*, following antibiotic exposure

Sophie MacVittie,[1] Saam Doroodian,[1] Aaron Alberto,[1] Maggie Sogin[1]

ABSTRACT   Microbial species that comprise host-associated microbiomes play an essential role in maintaining and mediating the health of plants and animals. While defining the role of individual or even complex communities is important toward quantifying the effect of the microbiome on host health, it is often challenging to develop causal studies that link microbial populations to changes in host fitness. Here, we investigated the impacts of reduced microbial load following antibiotic exposure on the fitness of the anemone, *Exaiptasia diaphana* and subsequent recovery of the host's microbiome. Anemones were exposed to two different types of antibiotic solutions for 3 weeks and subsequently held in sterilized seawater for a 3-week recovery period. Our results revealed that both antibiotic treatments reduced the overall microbial load during and up to 1 week post-treatment. The observed reduction in microbial load was coupled with reduced anemone biomass, halted asexual reproduction rates, and for one of the antibiotic treatments, the partial removal of the anemone's algal symbiont. Finally, our amplicon sequencing results of the 16S rRNA gene revealed that anemone bacterial composition only shifted in treated individuals during the recovery phase of the experiment, where we also observed a significant reduction in the overall diversity of the microbial community. Our work implies that the *E. diaphana's* microbiome contributes to host fitness and that the recovery of the host's microbiome following disturbance with antibiotics leads to a reduced, but stable microbial state.

IMPORTANCE   *Exaiptasia diaphana* is an emerging model used to define the cellular and molecular mechanisms of coral-algal symbioses. *E. diaphana* also houses a diverse microbiome, consisting of hundreds of microbial partners with undefined function. Here, we applied antibiotics to quantify the impact of microbiome removal on host fitness as well as define trajectories in microbiome recovery following disturbance. We showed that reduction of the microbiome leads to negative impacts on host fitness, and that the microbiome does not recover to its original composition while held under aseptic conditions. Rather the microbiome becomes less diverse, but more consistent across individuals. Our work is important because it suggests that anemone microbiomes play a role in maintaining host fitness, that they are susceptible to disturbance events, and that it is possible to generate gnotobiotic individuals that can be leveraged in microbiome manipulation studies to investigate the role of individual species on host health.

KEYWORDS   microbiome, Aiptasia, marine microbiology, antibiotic knockdown

Most animals rely on a complex microbiome to support individual fitness, health, and metabolism (1, 2). For example, microbiomes provide animals with essential nutrients, support reproductive pathways, and protect hosts from disease causing pathogens and toxic compounds (as reviewed in reference 3). An emerging hypothesis in microbiome research is that the unit of selection is indeed the "metaorganism," which is defined as the animal host together with its archaeal, bacterial, fungal, viral, and

Address correspondence to Maggie Sogin, esogin@ucmerced.edu.

The authors declare no conflict of interest.

See the funding table on p. 19.

microeukaryote associates (4). One challenge in understanding the role of individual microbial partners within the metaorganism is that for many systems we lack causal studies that support the intrinsic role of the microbiome in determining the phenotype of the host (but see references 5–7).

Marine cnidarians, such as jellyfish, sea anemones, and reef-building corals, host a broad diversity of microbial species within their tissue layers (8–11). Some of these microbial taxa play a fundamental role in supporting host health and metabolism. The most famous of which is the relationship between reef-building corals and their dinoflagellate, microalgal symbionts in the family *Symbiodiniaceae*. Overwhelming evidence demonstrates that the microalgal partner supports coral growth and survival through the transfer of sugars to the animal host (12, 13). In addition to *Symbiodiniaceae*, many cnidarians also require other microbial species (e.g., bacteria and fungi), to support complex metabolic pathways within the metaorganism (14). For example, cnidarian associated bacteria are likely involved in nitrogen and sulfur metabolism, and nutrient cycling (15–20), defense against host pathogens (5, 21–23), and, therefore, intrinsically supports coral health (21, 24, 25). Understanding the individual and collective role of these microbes in supporting cnidarians is critical as researchers and conservationists alike aim to develop new approaches (e.g., beneficial microorganisms for corals) for mitigating the impacts of environmental stress on coral reefs (3, 26, 27). Yet, much of the current work in defining microbial function of cnidarians remains correlative as there is a need to develop reductionist approaches that allow for the quantification of the impacts of individual microbial species or consortia on host health, metabolism, and fitness (28).

One promising approach in defining the role of individual microbes within metaorganisms is to rear hosts in the absence of their microbiome or with a reduced microbial load (5, 29). By generating these so-called gnotobiotic (i.e., known suite of microbial partners), or axenic (i.e., germ-free) systems, we can begin to dissect the individual roles of microbial species within the metaorganism and their impacts on cnidarian health. For example, the depletion of *Hydra's* microbiome revealed that microbial taxa were (i) critical in defending the animal against fungal pathogens (5), (ii) involved in cell signaling pathways that controlled host development (30), and (iii) regulated host physiology and phenotype (31, 32). In marine cnidarians, most of the microbiome depletion work is focused on quantifying the impacts of *Symbiodiniaceae* after removal from host tissues; however, other studies are beginning to quantify the reduction of other microbial species within the metaorganism (7, 29, 33–35). Here, we aimed to expand on these initial studies by (i) quantifying the impact of reducing the microbiome on cnidarian physiology and (ii) documenting patterns in microbiome recovery following disturbance using a longitudinal sampling approach. By quantifying shifts in cnidarian microbiomes through longitudinal sampling, we aim to describe community dynamics related to the recovery of cnidarian microbiomes. Defining how the microbiome recovers following disturbance is critical when assessing the efficacy and practicality of probiotic approaches in marine habitats. To meet these aims, we conducted a microbiome reduction experiment using the sea anemone, *Exaiptasia diaphana*, hereon referred to as Aiptasia.

Aiptasia is an emerging model system for exploring the role of microbial partners within marine cnidarians (36). Aiptasia houses the same type of *Symbiodiniaceae* partners within its gastrodermal tissue layer as reef building corals. Consequently, much of the current work in Aiptasia is focused on studying shifts in the cell and molecular machinery following the removal of the algal symbiont (i.e., bleaching)(37–39). However, like corals, Aiptasia also hosts a complex core microbiome, likely consisting of 24–44 bacterial species with hundreds to thousands of accessory members (as defined by amplicon sequence variants or operational taxonomic units) that live in the anemone's tissue and mucus layers (10, 34, 40, 41). It is possible to reduce the microbial load of the Aiptasia microbiome using antibiotics (33), however microbial depletion is unstable in the presence of biofilms (34), and thus it is difficult to maintain in anemone cultures. Importantly, it is not clear how the depletion of the microbiome directly impacts overall

Aiptasia physiology and health. To begin to disentangle the roles of individual microbes within Aiptasia, it is essential to quantify how the microbial community shifts during depletion and what the overall impact is on the metaorganism.

In this study, we set out to quantify the effects of antibiotic treatment on the bacterial load and composition of the Aiptasia microbiome while testing to determine if our experimental treatments impacted metaorganism fitness. Leveraging lessons learned from past knockdown studies (33, 34), we exposed individual sea anemones to two different antibiotic solutions for 3 weeks and monitored the impact on the microbiome (i.e., both alpha and beta diversity) and physiology (i.e., total biomass, algal cell density, and asexual reproduction rate) during recovery. Overall, our results revealed that exposure to antibiotics reduced the abundance of the Aiptasia bacterial community, shifted the composition of community members, which resulted in fitness declines, as measured in total biomass, algal cell density, and asexual reproduction metrics.

## RESULTS

### Experimental overview

We set out to quantify the impacts of antibiotic exposure on the disturbance and recovery of the Aiptasia (host strain H2 with *Breviolum minutum*) microbiome and associated host fitness. The 76-day-long experiment was divided into three phases: priming, treatment, and recovery (Fig. 1). During the priming, anemones that were treated with antibiotic solutions during the treatment phase were first held in filtered artificial seawater (FASW) for 33 days to reduce overall microbial load. After which, these anemones were treated with either antibiotic solution 1 (ABS1; 50 µg/mL of carbenicillin, chloramphenicol, nalidixic acid, and rifampicin) or antibiotic solution 2 (ABS2; 50 µg/mL of neomycin, penicillin, rifampicin, and streptomycin) for a total of 22 days. We chose to use two different solutions in order to (i) determine if one solution is more effective than the other, (ii) compare differences in microbial community composition as the different types of antibiotics have varying mechanisms of action to remove microbial species, and (iii) because both solutions are known to deplete cnidarian microbiomes (33, 42). During recovery, treated anemones were held in FASW for 21 days. The timing of the experimental phases was based on prior work in anemones showing that priming in FASW for 33 days, followed by a 22-day antibiotic exposure is successful in knocking

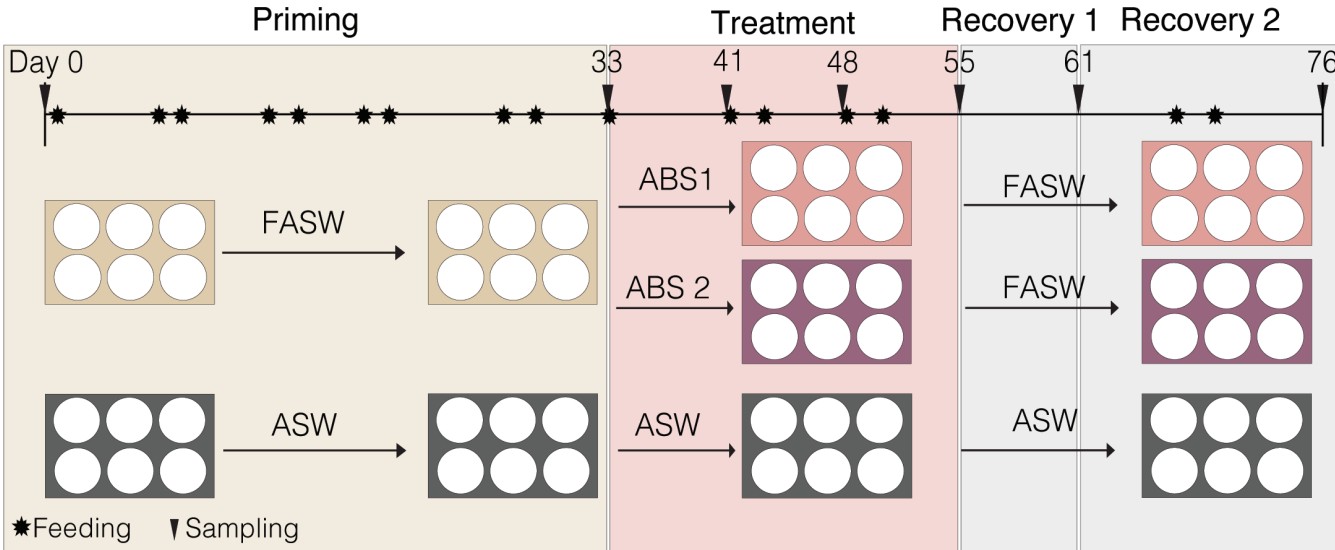

**FIG 1** Experimental overview outlining the steps in the 76-day experiment. First, individual Aiptasia polyps were held in either artificial seawater (ASW; control) or filtered artificial seawater (FASW) for 33 days. Anemones in FASW were then treated with either ABS1 or ABS2 solutions for 22 days and subsequently allowed to recover in FASW for 21 days. Anemones were sampled (black triangle) at least four days post feeding to avoid sampling shifts in microbiomes related to animal husbandry (black star).

down the Aiptasia microbiome (33, 34, 43).We chose to follow the anemones for 3 weeks during recovery to determine the point at which the microbiome recovered to control levels. Throughout the experiment, we sampled individual anemones from the treatment and control groups (i.e., individuals held in artificial seawater [ASW]) on days 0 (baseline), 33 (priming), 55 (treatment), 61 (recovery 1), and 76 (recovery 2) for amplicon sequencing using the V4 region of the 16S SSU of the rRNA gene, bacterial load, biomass, *Symbiodiniaceae* density, and asexual reproduction rate (Fig. 1).

## Exposure to antibiotics depletes Aipstasia's bacterial load during treatment

Anemones treated with both ABS1 and ABS2 solutions had a significantly reduced bacterial load, as measured by colony forming unit counts (CFUs), during treatment and up to 1 week of recovery in FASW (Kruskal-Wallis $P < 0.001$; Fig. 2A; Tables S1 and S2). Unlike previous studies (34, 43), there were no differences ($P > 0.05$) in CFU counts between anemones held in FASW (average CFU cells/mL = $8.1 \times 10^4 \pm 2.3 \times 10^4$) in comparison to the controls (average CFU cells/mL = $1.2 \times 10^5 \pm 8.9 \times 10^4$) during priming. Rather, CFU counts only significantly decreased ($P < 0.05$) in anemones exposed to the ABS solutions during treatment (days 41 and 55; average CFU cells/mL = $32 \pm 2.5$ to $504 \pm 4.2$) in comparison to the controls (average CFU cells/mL = $1.1 \times 10^6 \pm 8.3 \times 10^5$ to $7.2 \times 10^6 \pm 6.7 \times 10^6$). During recovery in FASW, CFU counts remained significantly lower ($P < 0.05$) in treated anemones for up to 1 week post-treatment (day 61; average CFU cells/mL = $336 \pm 12$ to $5.2 \times 10^4 \pm 5.1 \times 10^4$) in comparison to the control (average CFU cells/mL = $5.9 \times 10^6 \pm 5.2 \times 10^5$). However, by 3- weeks post-treatment (day 76), CFU counts returned to similar levels as the control group (average CFU cells/mL = $7.3 \times 10^5 \pm 5.3 \times 10^5$ to $4.2 \times 10^6 \pm 2.5 \times 10^6$). Throughout the experiment, there were no significant differences ($P > 0.05$) in CFU counts between anemones held in ABS1 versus ABS2 solutions (Table S1).

## Exposure to antibiotics reduced anemone fitness

Anemones treated with ABS1 and ABS2 had reduced total protein content in comparison to the controls during treatment and recovery, suggesting our treatment regimes led to significant decreases in overall biomass (Fig. 2B; Tables S1 and S2). While total protein content in anemones held in FASW (average µg/mL of protein = 295.13 ± 79.90) was decreased in comparison to the control (average µg/mL of protein = 487.95 ± 47.94), differences between treatments was not significant. Total protein content was significantly reduced ($P < 0.01$) after only 7 days of exposure in ABS1 (average µg/mL of protein = 212.60 ± 47.25) compared to the control (average µg/mL of protein = 583.98 ± 67.06), whereas ABS2 was slightly reduced but not significantly different from the control (average µg/mL of protein = 372.01 ± 76.09). Total protein content was significantly ($P < 0.05$) reduced in ABS1 and ABS2 throughout the remainder of treatment (Tables S1 and S2). During recovery at day 61, anemones continued to diminish in size in both ABS1 (average µg/mL of protein = 109.46 ± 29.53) and ABS2 (average µg/mL of protein = 206.86 ± 36.02), while the control maintained their size (average µg/mL of protein = 677.62 ± 52.78). At the final recovery time point (76 days), treated anemones began to increase in biomass (average µg/mL of protein = 168.36 ± 24.31 to 274.79 ± 56.18), which was no longer significantly different from the control (average µg/mL of protein = 409.12 ± 42.06).

Anemones treated with antibiotics had reduced asexual reproduction rate, as measured by pedal laceration, during treatment (Kruskal-Wallis test $P < 0.0001$; Fig. 2C; Tables S1 and S2). FASW treatment during priming did not impact on the rate of pedal laceration ($P > 0.05$; mean pedal laceration rate = 0.26 ± 0.11 individuals week$^{-1}$). During recovery, anemones held in ABS1 did not pedal lacerate even during the final sampling time point, while anemones held in ABS2 began to pedal lacerate at 71 days (mean pedal laceration rate = 0.60 ± 0.40 individuals week$^{-1}$). Both rates were significantly lower than the control (mean pedal laceration rate = 3.5 ± 0.92 individuals week$^{-1}$).

Symbiont density did not vary significantly in individuals treated with ABS2 or FASW (two-way ANOVA, $P > 0.1$, Fig. 2D; Tables S1 and S2). In contrast, symbiont density

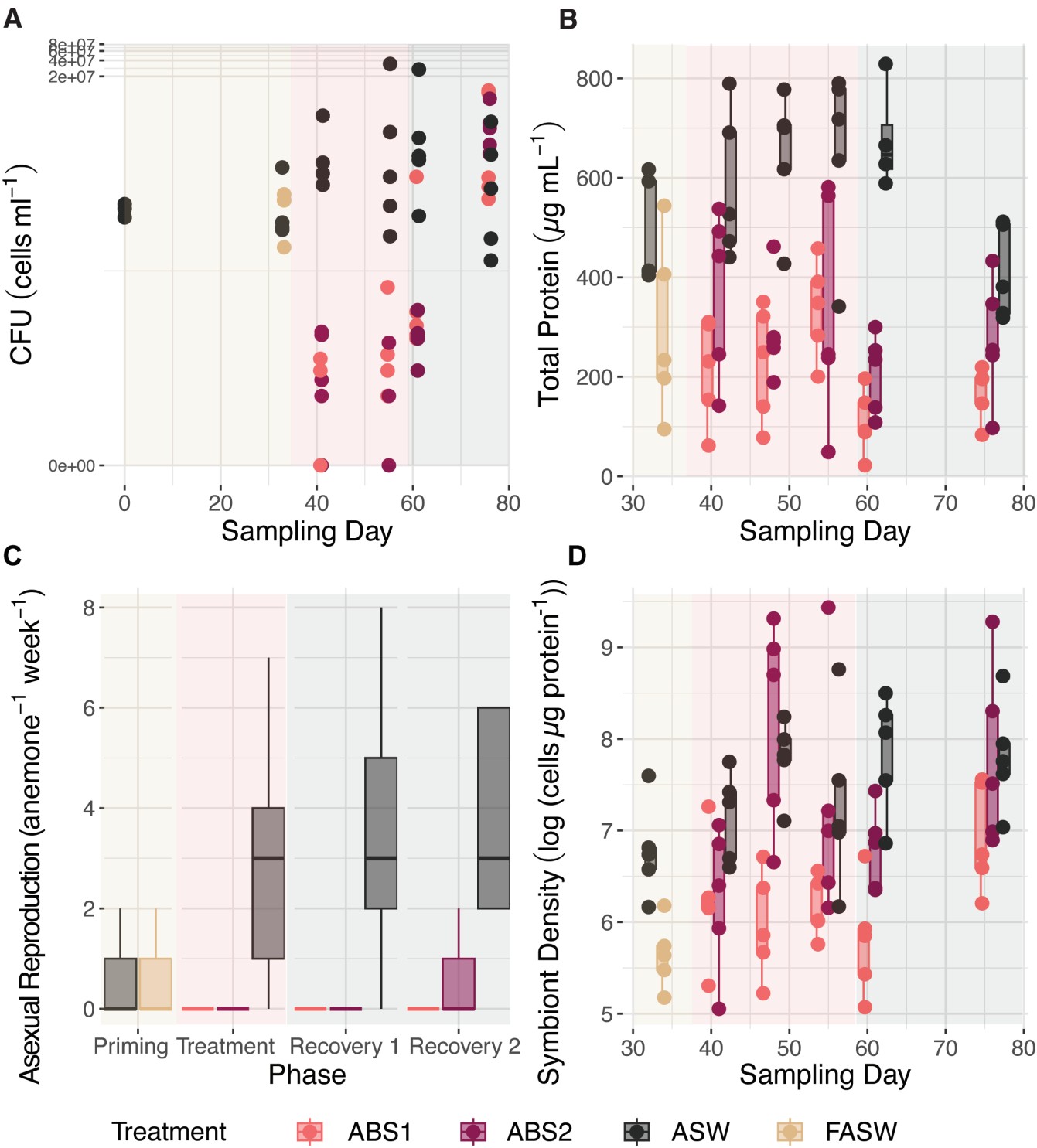

**FIG 2** Antibiotic exposure reduced bacterial load and host fitness. (A) Colony forming unit counts (CFUs, cells mL$^{-1}$) significantly differed across treatments and sampled time points (two-way ANOVA $P < 0.001$). Aiptasia exposed to antibiotics experienced a significant (Tukey HSD adjusted $P < 0.001$) reduction in bacterial load during treatment on days 41 and 55, and during the first phase of recovery on day 61. CFU counts returned to control levels on day 76 (day 0: $n = 3$, otherwise $n = 5$; day 48: CFU counts were excluded due to plate contamination). (B) Total protein concentrations differed significantly across time (two-way ANOVA $P = 0.01$) and experimental treatments (two-way ANOVA $P < 0.001$) between treated anemones and the control group. Boxplots of total protein (µg mL$^{-1}$) measured from each polyp ($n = 5$), revealed that anemone biomass significantly declined (Tukey HSD adjusted $P < 0.001$) initially during the priming phase when individuals were held in FASW in comparison to the control and remained significantly reduced in concentration in comparison to the control

**FIG 2** (Continued)

throughout the remainder of the experiment (Tukey HSD adjusted $P < 0.001$). (C) Exposure to antibiotics significantly reduced the asexual reproduction rate of individual anemones ($n = 6$) during treatment and recovery (Kruskal-Wallis $P < 0.001$). Boxplots of asexual reproduction rate as measured by number of observed pedal lacerates per individual per week showed that both antibiotics halted pedal laceration rate until the final recovery time point, when anemones held in ABS2 started to recover their capacity to undergo asexual reproduction. (D) Algal cell densities (cell density µg protein$^{-1}$) differed significantly across time and experimental treatments between treated anemones and the control group ($n = 5$) (two-way ANOVA $P < 0.001$). Boxplots of normalized symbiont density show that FASW reduced algal cell density during the priming phase. While anemones in ABS2 recovered their algal populations during treatment, anemones held in ABS1 had reduced algal population until the final recovery period. See Table S1 for all model statistics and Table S2 for means and standard errors.

decreased in anemones treated with ABS1 during treatment and in the first week of recovery. Two weeks into treatment (day 48), symbiont density in ABS1 (average cell/µg protein = 447.10 ± 114.63) was significantly lower ($P < 0.05$) compared to the control (average cell/µg protein = 2,570.09 ± 419.78). After 1 week of recovery, anemones in ABS1 were visually lighter and had a significantly lower ($P < 0.01$) symbiont density (average cell/µg protein = 388.21 ± 117.10) when compared to the control (average cell/µg protein = 2,963.21 ± 700.96). By the end of the recovery period, symbiont density in ABS1 (average cell/µg protein = 1,167.09 ± 297.94) returned to control levels (average cell/µg protein = 2,851.40 ± 814.97). While there were no discernible patterns in ABS2 samples, Aiptasia exposed to ABS2 had much higher variability in protein content than all other treatments throughout the experiment.

## Exposure to antibiotics shifts the composition of the core Aiptasia microbiome

To quantify the impacts of antibiotic exposure on the Aiptasia microbiome, we sequenced the V4 region of the 16S rRNA gene from individual anemones. In total, we recovered between 83 and 890,282 reads from each sample. Samples and extraction blanks with fewer than 20,000 or greater than 250,000 reads were removed from the analysis. Removed samples included four individuals treated with ABS solutions, one control sample, two filtered seawater samples, and one extraction blank. After quality filtering the read set, determining amplicon sequence variants (ASVs), and removal of contaminating, ribosomal, and chromosomal ASVs, the resulting data set contained 412 individual taxa across 82 samples. To determine how antibiotics impacted the Aiptasia microbiome, we first compared alpha diversity across treatments and sampling time points (days 0, 33, 55, 61, and 76; Fig. 3A). We then calculated the core (Fig. 3D) membership based on prevalence and abundance of ASVs from the control samples. Finally, we assessed shifts in beta-diversity of the core (Fig. 3B and C) across treatments and sampling time points (days 0, 33, 55, 61, and 76).

We observed significant differences in alpha diversity during priming and recovery prior to and following antibiotic exposure (Fig. 3A). There was a significant interaction between treatment and sampling time ($P < 0.0000367$) on the overall number of ASVs per individual. The number of ASVs in the control samples did not significantly change over time (average no. ASVs ± s.e. =105.25 ± 0.6 to 75.8 ± 2.4). There was also no significant difference in the number of ASVs in either of the ABS treatments during treatment (day 55, average no. ASVs ± s.e. =105 ± 6.1 to 86.6 ± 7.6). However, we did observe a significant decrease in the observed number of ASVs in anemones treated with ABS1 on days 61 and 76 (average no. ASVs ± s.e. =52 ± 13.9 to 51.8 ± 7.6) and anemones treated with ABS2 on day 76 (average no. ASVs ± s.e. =43.6 ± 6.9). Indeed, the average number of ASVs per individual in anemones treated with ABS solutions was half that observed in the control samples (average no. ASVs ± s.e. =867.4 ± 5.8) at the final sampling time point.

We performed a core microbiome analysis using anemones only exposed to ASW (control group). We chose to include only the control samples in our core analysis to determine how our experimental treatments impacted the composition of the core microbiome throughout the experiment. The core, as defined by taxa present in at least 90% of the control samples, consisted of 51 ASVs (Fig. 3D). The core ASVs included taxa

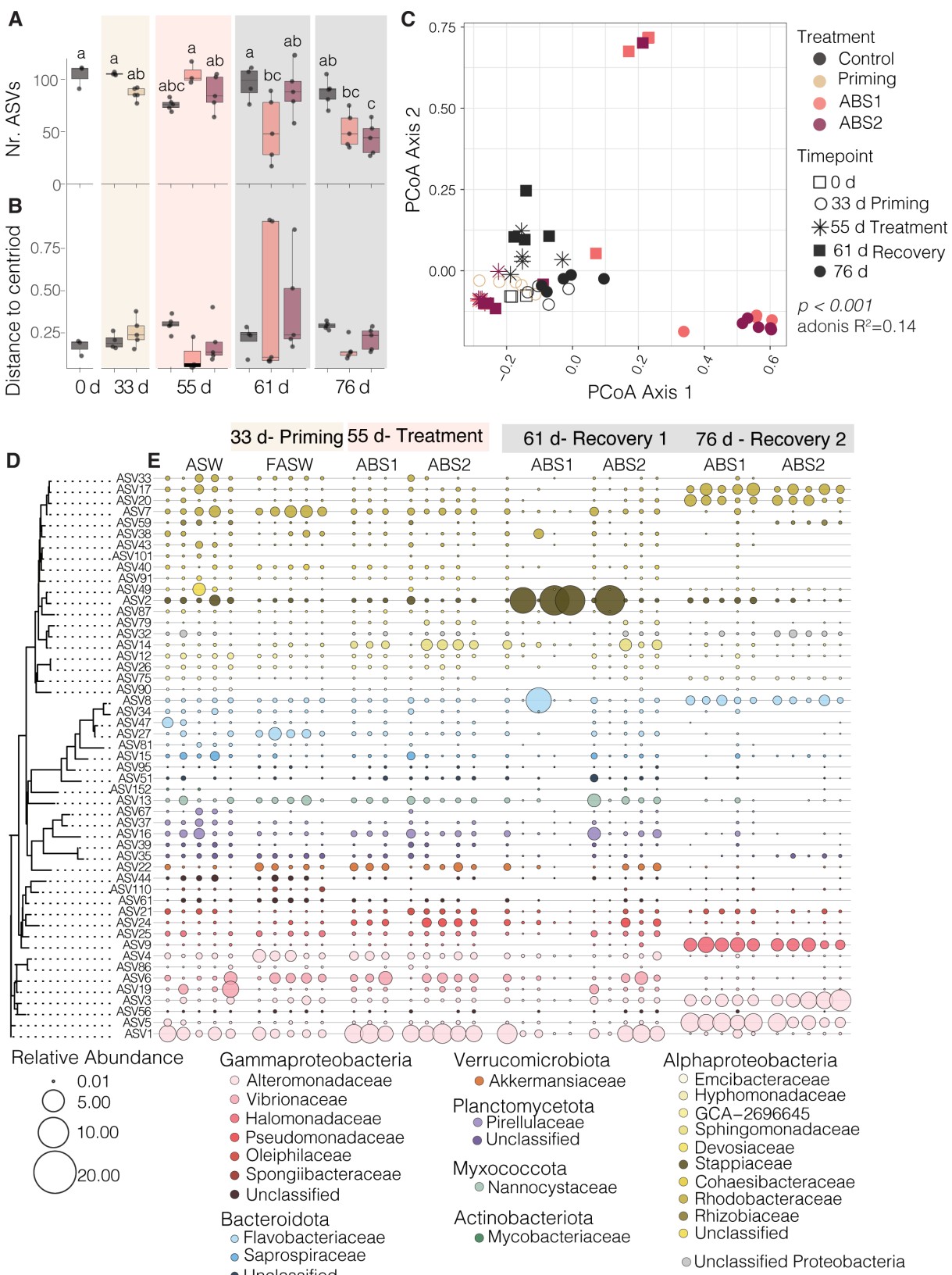

**FIG 3** Antibiotic exposure shifts the core microbiome of Aiptasia. (A) Boxplots showing alpha diversity, as reported as the number of observed ASVs, revealed that anemones exposed to antibiotics had a significantly less diverse community in comparison to the controls ($n$ = 4–5 per treatment per time point)(two-way ANOVA $P < 0.001$). (B) Boxplots of beta-dispersion values and (C) a PCoA analysis calculated from a Bray-Curtis distance matrix of the core (D) microbiome (Continued on next page)

**FIG 3** (Continued)

revealed that beta-diversity significantly differs between treatment groups and across sampling time points (ANOSIMS $R^2$ = 0.14, $P$ < 0.004). In panel A, the letter represents results from a Tukey's Honest Significant Difference test, where groups that are connected by the sample letter are not statistically different. (D) A dendrogram of the core microbiome based on sequence distances grouped taxa according to phylogenetic grouping. (E) Bubble plots showing shifts in the relative abundance of the core members across treatment groups and sampling time points. Only a subset of the ASW control samples collected at each sampling time point are displayed. All treatment samples are ordered by sampling time point and treatment conditions. The size of the bubble corresponds to the relative abundance of the ASV. If there is no point shown in the graph, the ASV was not detected in the sample. ASW = artificial seawater, FASW = filtered artificial seawater, ABS = antibiotic solution. Results of statistical tests are reported in Table S1 and the core community is identified in Table S3.

belonging to the bacterial classes Actinomycetia ($n$ = 1), Bacteroidia ($n$ = 8), Polyangia ($n$ = 1), Phycisphaerae ($n$ = 1), Planctomycetes ($n$ = 3), Alphaproteobacteria ($n$ = 19), Gammaproteobacteria ($n$ = 15), Verrucomicrobia ($n$ = 1), and unclassified Proteobacteria ($n$ = 1; Table S3). The core community represents between 41% and 99% of each sample's relative abundance (Fig. 4), indicating that we are capturing the majority of the community for most samples across treatment groups.

Our results revealed that the core microbiome differed significantly following exposure to antibiotics. Using a principal coordinate analysis (PCoA) analysis based on

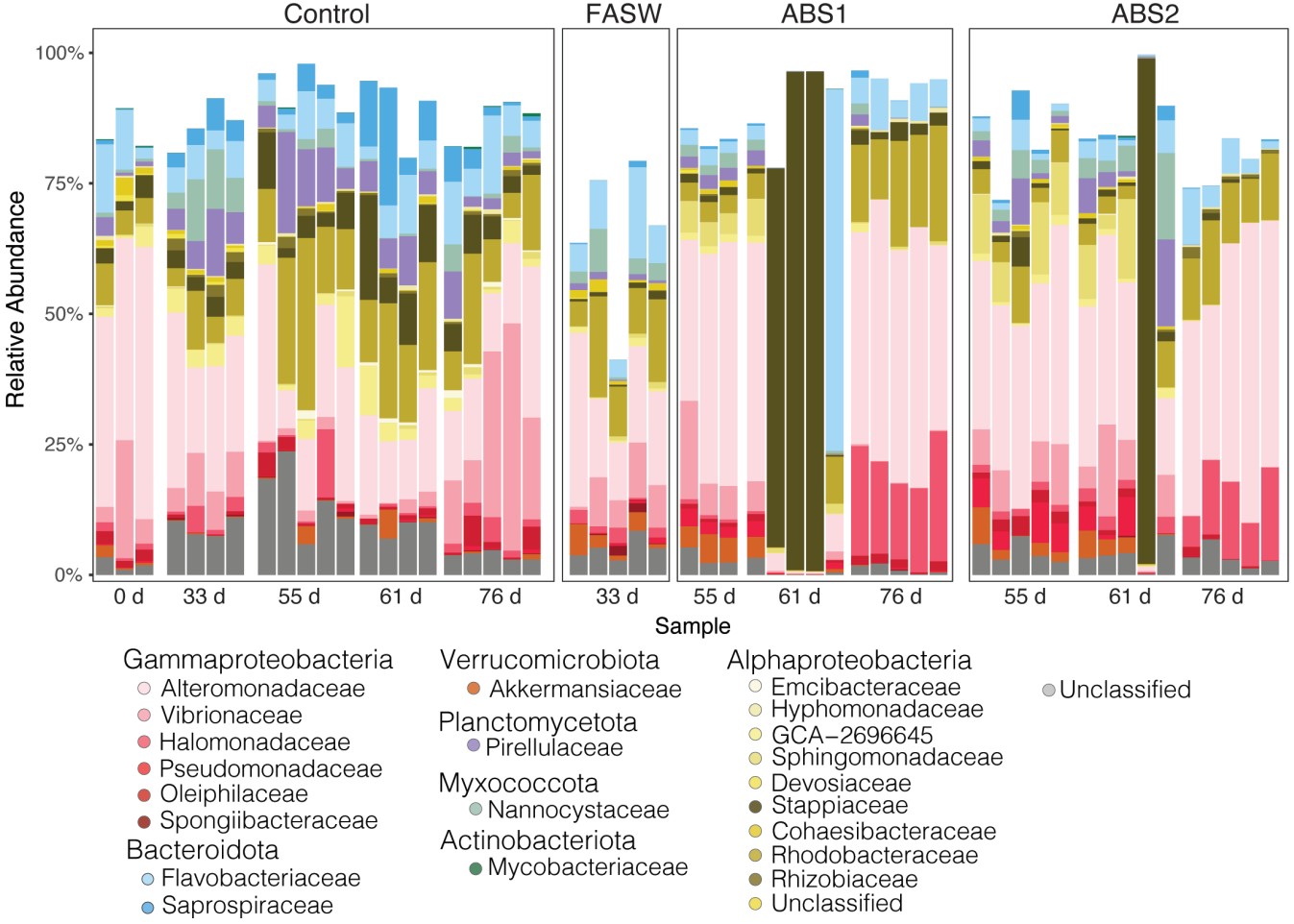

**FIG 4** The Aiptasia core microbiome across treatment conditions. The bar plot revealed the relative abundance of each of the 51 ASVs that make up the core microbiome of the non-treated anemones. Members of the core microbiome were removed following exposure to FASW and both antibiotic solutions. At the final recovery time point (day 76), the microbiome became more consistent across individual anemones. ASVs are grouped and colored according to family level classification, each bar represents a single sample grouped according to treatment and sampling time point prior to (0 and 33 days) and directly following antibiotic exposure (55 days) and at both recovery time points (55 and 76 days). ASW = artificial seawater, FASW = filtered artificial seawater, ABS1 = antibiotic solution 1, ABS2 = antibiotic solution 2.

a Bray-Curtis distance matrix, we showed that the composition of the Aiptasia core shifted as a function of antibiotic exposure and sampling time point. A permutation multivariate ANOVA (PERMANOVA) test found a weak, but significant interaction of treatment and sampling time point on multivariate microbiome composition (adonis $R^2$ = 0.14, $P$ < 0.001; Fig. 3C). A subsequent beta-dispersion analysis showed that there were no significant differences between group dispersions ($P$ = 0.271), or the distance to the group's centroid. These results suggest that there is a strong separation in beta-diversity between treatments, with the lowest beta-dispersion values for anemones treated with ABS1 solution sampled during the recovery time point (Fig. 3B).

Of the 51 core members of the Aiptasia microbiome, 31 were removed either directly after ABS exposure or were absent from the microbiome in treated anemones at either of the two recovery time points (Fig. 3E; Table S3). Of the seven taxa that were removed directly after ABS treatments, all ASVs, except for one species of Rhizobiales, remained knocked down during recovery. Knocked down taxa included the Actinomycetia species, *Dietzia psychralacliphila*, a species of Flavobacteriales, *Muricauda* sp. 004804315, the Gammaproteobacteria, *Spongiibacter tropicus*, and unknown species belonging to the Planctomycetes, Alphaproteobacteria, and Gammaproteobacteria. Intriguingly, 23 of the 31 species that were ultimately removed from the Aiptasia core, were still present directly following ABS treatment. However, these taxa were lost or significantly reduced in the core community during recovery, which led to the ultimate reduction in bacterial diversity at the end of our experiment. Bacterial species that were removed included the Gammaproteobacteria *Oceanospirillum linum*, and the Alphaproteobacteria groups *Ruegeria* sp., *Emicbacter* sp., and *Cohaesibacter* sp, and species within the Bacteroidia, Phycisphaerae, Planctomycetes, Alphaproteobacteria, Gammaproteobacteria, and Verrucomicrobiales (Fig. 3E; Table S3).

## Differential patterns of microbial resistance, susceptibility, and selection by antibiotic exposure

Exposure to antibiotics differentially impacted the presence and absence of specific ASVs within the Aiptasia microbiome. We used the ALDEx2 package in R to first calculate the center log ratio to determine the within-sample, geometric mean of the read counts for each ASV (44, 45). We then used ALDEx2 to identify individual ASVs that were significantly different (BH adjusted $P$ < 0.05) between the ABS treatment and control group and had an ALDEx effect size greater than 2 or less than −2. For our analyses, we made two decisions to help guide the interpretation of our results. First, we chose to determine the differentially abundant (DA) ASVs for the entire data set, rather than focus on the core community, which allowed us to identify all taxa that were impacted by the antibiotic exposure. We also chose to calculate the differential abundance of each ASV between the individual ABS treatment groups and the control sampled at days 55, 61, and 76 to determine how each antibiotic treatment impacted the microbiome during treatment and recovery.

In total, we identified 37 unique ASVs that were differentially abundant across all comparisons (Fig. 5 and 6). In our analyses, there were far more DA-ASVs identified during the final recovery time point (76 days) for both ABS1 (9 total; Fig. 5C) and ABS2 treatments (29 total; Fig. 5C). In contrast, we only observed four DA-ASVs in anemones collected directly after ABS1 exposure (Fig. 5A). We did not observe any DA-ASVs in anemones collected directly after ABS2 treatment (Fig. 5D). Of the differentially abundant ASVs, 65% were members of the Aiptasia core microbiome (Table S4).

Next, we plotted the relative abundance of the 37 DA taxa across the three time points (days 55, 61, and 76) for both ABS treatments and control conditions (Fig. 6). To determine how each DA taxa is impacted by ABS exposure, we classified each ASV based on their shift in relative abundance directly after treatment and during recovery. ASVs were classified into one of four categories: (i) susceptible, (ii) resistant, (iii) selected, and (iv) opportunistic.

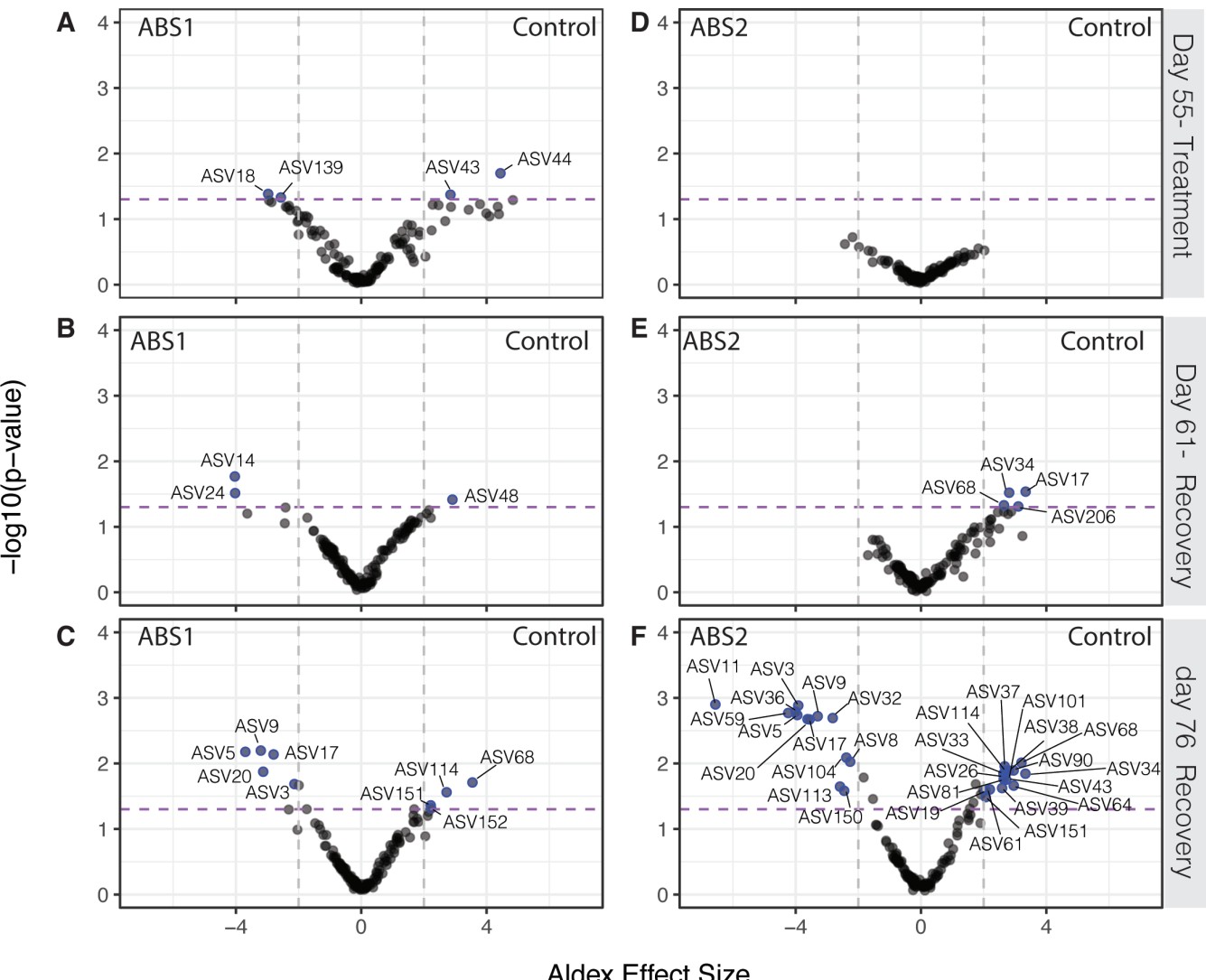

**FIG 5** Exposure to antibiotics results in differential abundance of select ASVs one and two weeks after recovery. Volcano plot of all ASVs shows differentially abundant taxa (BH adjusted Welch's *t* test *P* < 0.05) between (A–C) ABS1 and control groups and (D–F) ABS2 and control groups (A and F) directly following ABS treatment (day 55), (B and E) one week post-ABS treatment (day 61) and (C and F) three week (day 76) post-ABS treatment. Taxa that are significantly different between treatment groups (BH adjusted *P* < 0.05) and a calculated ALDEx effect size >2 or <−2 are represented by blue points and labeled with the ASV number. The purple dashed line represents a BH adjusted, Welch's *t* test *P* < 0.05. The gray dotted lines are the threshold for the ALDEx effect size >2 or <−2. ASVs that are enriched in the ABS treatments are plotted on the left side of the graph, while ASVs enriched in the control are plotted on the right.

Susceptible taxa included ASVs that were either completely removed or significantly reduced in anemones treated with antibiotics directly following treatment (day 55). Susceptible taxa either remained reduced in relative abundance (e.g., ABS 44 in Fig. 6A), were eliminated during recovery (e.g., ASV 34 in Fig. 6B), or were removed during treatment but recovered to similar relative abundances as observed in the control group during recovery (e.g., ASV 104 in Fig. 6B). In total, we identified eight ASVs in ABS1 and 12 ASVs in ABS2 as susceptible to the antibiotics.

Taxa that were resistant to the antibiotics in the DA analysis included ASVs that had relative abundances at similar levels to the control group directly following antibiotic treatment, but either had significantly higher or lower relative abundances during recovery. For example, some taxa were not directly impacted by the ABS exposure during treatment, but during recovery they were removed from the microbiome (e.g., ASV5 in Fig. 6A). Conversely, other taxa that were not removed during exposure, ended up

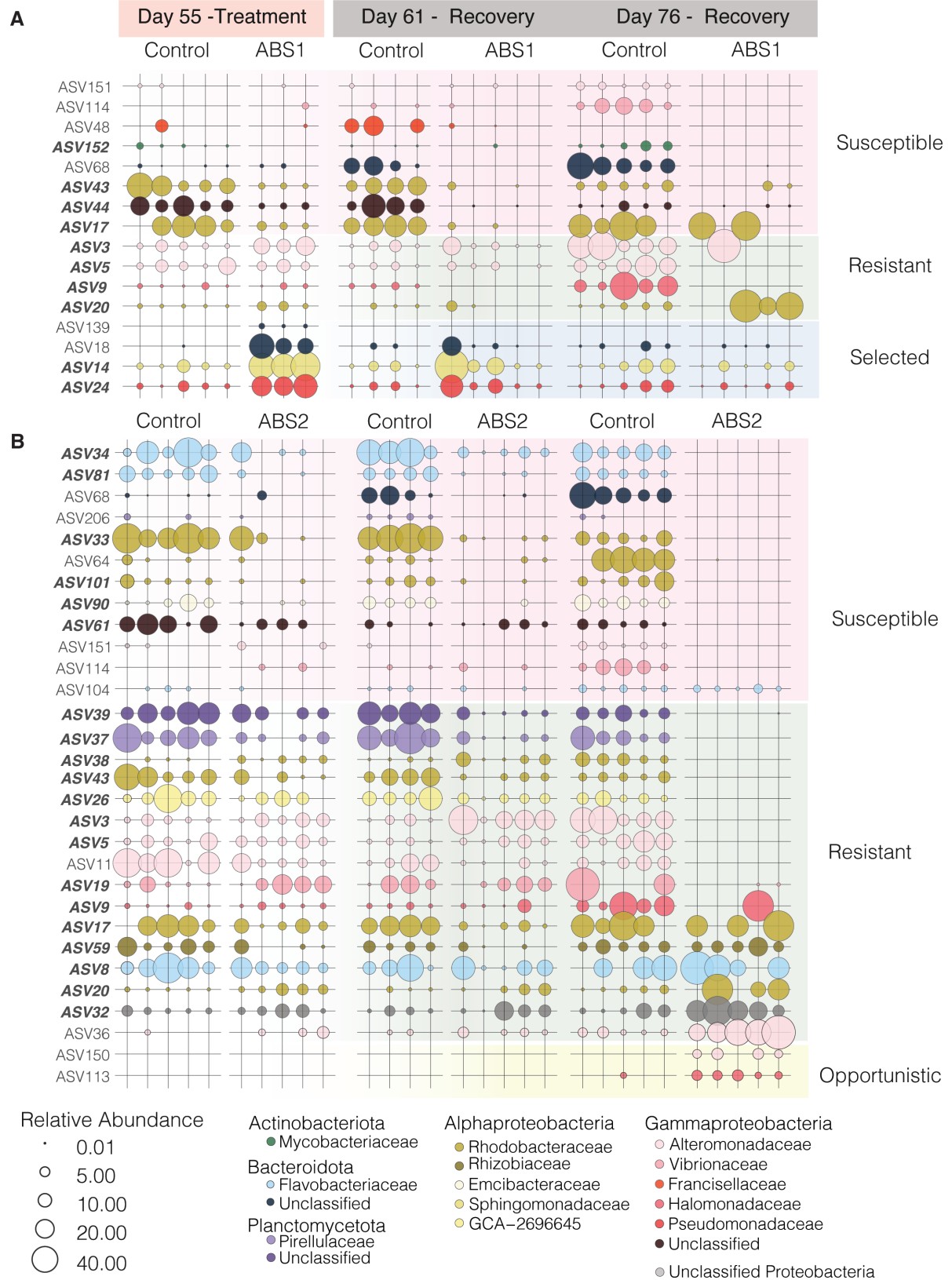

**FIG 6** Relative abundance of differentially abundant taxa shows the ASVs were selectively knocked down during ABS exposure and remained depleted during recovery. The relative abundance of ASVs determined to be differentially abundant according to the ALDEx2 analysis (see Fig. S2) are plotted for (A) ABS1 and (B) ABS2. The size of each point represents the relative abundance of that taxa within the selected sample. Points are colored according to family-level

**FIG 6** (Continued)

classification. ASVs lacking a point represent the absence of that ASV within the sample. Row shading is used to classify taxa based on their shifts in relative abundance throughout the experiment. Taxa were classified as susceptible or resistant to antibiotics, selected for by the antibiotics or were considered opportunistic taxa.

dominating the community after three weeks of recovery (day 76) in FASW. Naturally, our DA analysis did not identify taxa that were resistant to ABS but did not change in relative abundance throughout the experiment (see Fig. 3).

Finally, we also identified DA taxa that we considered to be selected for by the antibiotic exposure or were present in the microbiome as an opportunist. ASVs that had relative abundances that followed these patterns included taxa that were present in the microbiome of anemones that were treated with ABS solutions but were either lower in abundance or not present in the control samples. In total, we identified four taxa that were selected by the antibiotics in our ABS1 treatment. Selected taxa had higher relative abundances in individuals treated with ABS solutions in comparison to the control directly after treatment (55 days; Fig. 6A), but the taxa were either removed (e.g., ASV139) or reduced to control levels during the recovery period (e.g., by 76 days). While there were no taxa that were selected by the ABS2 treatment, there were two taxa (ASV150 and ASV113) that we classified as opportunistic (Fig. 6B). These two taxa only appeared in anemones treated with ABS2 after three weeks of recovery in FASW.

## DISCUSSION

### Antibiotic exposure reduced bacterial load up to one week post-treatment

In our study, we showed that antibiotic solutions decreased the microbial load and composition of the Aiptasia microbiome. It is likely that our colony forming unit assay, which allowed us to show that bacterial load is reduced in treated anemones, missed key bacterial groups that remain refectory to cultivation on marine agar. However, our results are consistent with previous work in cnidarians which showed that antibiotics reduced the abundance of bacteria within the host microbiome following 5–7 days of exposure (33, 35, 46, 47). Surprisingly, our community composition analysis revealed that the microbiome only shifted in treated anemones during recovery. Indeed, by the final sampling time point (76 days), treated anemones were less diverse. We were also able to stably remove 29 of the 51 core members of the microbiome. Our results extend past studies seeking to manipulate and reduce the Aiptasia microbiome by demonstrating that it is indeed possible to use antibiotics to reduce the complexity of the community, maintain low microbial loads for up to one week post-treatment, and generate a new microbial stable state within the host.

In parallel to reducing the microbial load, we also observed that antibiotic exposure reduced host fitness, as measured by total biomass, *Symbiodinaceae* densities, and asexual reproduction rates. Because antibiotics are known to also impose metabolic consequences on eukaryotic cells (48), it is challenging to definitively link reduction in biomass, asexual reproduction rate, and *Symbiodinaceae* densities to a reduction in microbial load. However, some interesting patterns emerged in our physiological data.

Anemones treated with both ABS solutions overall had lower biomass throughout the experiment in comparison to the controls, except for anemones that had been treated with ABS2 where biomass recovered to control levels at the final sampling time point. Apparent recovery of biomass could also be interpreted as a decrease in control anemone biomass due to reduced feeding at the end of the experiment. Another possibility is that anemones treated with both ABS solutions were unable to feed on sterile brine shrimp throughout treatment leading to a reduction in overall biomass. This matches observations that treated anemones did not expel visible food pellets following feeding. Our results are consistent with similar observations from Hydra, namely hydra polyps held in antibiotic solutions were unable to feed on *Artemia*, however regained their feeding phenotype once transferred to fresh media (49). We observe a similar

behavior in Aiptasia, however antibiotic exposure in our anemones may have more serious phenotypic consequences than their freshwater relatives as Aiptasia seems to halt or significantly reduce their feeding behavior even when transferred to antibiotic-free media. Coupled with a reduction in biomass, anemones held in ABS solutions were not able to reproduce asexually during treatment. It is unlikely that the reduction in biomass alone led to a reduction in pedal laceration rates during antibiotic exposure as anemone growth and pedal laceration rates are not correlated (50–52). Furthermore, starved anemones tend to have increased asexual reproduction rates compared to fed individuals (50, 51). In the freshwater cnidarian, *Hydra*, antibiotic exposure also led to a reduction in anemone budding rates (42). Our result supports the hypothesis that the presence of bacterial partners plays a critical role in the asexual reproduction of all anemones. Finally, our study revealed that antibiotic exposure also led to reduced *Symbiodiniaceae* densities in anemones treated with ABS1 solutions, but not with ABS2. Because we show differences in symbiont densities as a function of type of antibiotics two possibilities emerge. Either our ABS1 solution had a direct negative impact on the *Symbiodiniaceae* symbiont itself, or ABS1 differentially removes a key bacterial partner of the algae that is not targeted by ABS2 and resulted in the differential reduction of the algal population. Either way, our study extends past studies by quantifying the physiological impacts of antibiotic treatments on host fitness and suggests that bacterial load and composition may play an important role in the maintenance of Aiptasia fitness.

## Antibiotic exposure resulted in a less diverse, more consistent microbial community

Antibiotic exposure resulted in anemones with reduced complexity of the microbial community. Leveraging recommendations made by previous studies (33, 34), we successfully applied two different types of antibiotic solutions to individual anemones and observed a reduction in bacterial load and alpha diversity. While bacterial reduction only lasted one week post-ABS exposure, the composition was significantly shifted throughout recovery. We did not test the stability of the post-ABS community past three weeks in FASW, but it is possible that the composition would have remained stable considering the anemones were held in sterile seawater and only fed antibiotic-treated *Artemia* throughout the experiment. We suspected that we limited the introduction of new environmental bacteria to the host system, which enabled the long-term maintenance of the reduced microbial state.

Our work indicates that the recovery period following antibiotic exposure is critical for the establishment of a less complex microbiome model for Aiptasia. Recovery allows for the stabilization of the microbial communities and is likely impacted by the members of the microbiome that are differentially removed versus retained. Interestingly, not all cnidarians or marine invertebrates can maintain a low-diversity microbiome after antibiotic exposure. For example, the temperate coral *Astrangia poculata* recovered their microbial communities after only two weeks of recovery following exposure to antibiotics (35) and the Aiptasia strain, AIMS2, never showed a reduction in microbial diversity during or following antibiotic exposure (34). Similarly, antibiotic treatments, albeit much shorter than our three-week exposure, of the marine sponge, *Halichondria panciea*, resulted in a more diverse microbial community than control samples during recovery and ultimately lead to dysbiosis of the community (53). In *Acropora muricata*, bacterial communities failed to recover for at least four days post-antibiotic treatment (46). Variations in organismal responses across cnidarian groups may be due to variations in the production of antimicrobial compounds within the surface mucus layers (54). Our work highlights the need for additional longitudinal studies showing the recovery dynamics of cnidarian microbiomes following antibiotic or other disturbance to better understand the role of individual taxa in microbiome recovery.

While we were able to generate and maintain a lower diversity microbial community, we were not successful in completely removing the Aiptasia microbiome to generate axenic, or microbe-free, hosts. It is likely that our antibiotic treatments were not effective

at reaching intracellular niches where microbial taxa might reside (e.g., those that live in symbiosis with *Symbiodiniaceae*) or that the Aiptasia host harbors antibiotic-resistant bacteria that are challenging to remove with current drugs. Indeed, two of the ASVs in our study that were either not removed during antibiotic exposure or were considered resistant taxa were closely related to bacterial species that were either shown to be resistant to a full suite of antibiotic drugs (e.g., *Marisediminitalea aggregate* [55]) or contained antibiotic genes within their genomes (e.g., *Aureliella hegolandensis* [56]). One option, that until recently was not available as the life cycle of Aiptasia was not closed in the laboratory (57), is to apply antibiotics to newly settled Aiptasia larvae that lack their intracellular symbionts. Similar approaches that work to treat animals with antibiotics early in their life cycle have successfully resulted in axenic strains of *Drosophilia melanogaster*, *Caenorhabditis elegans*, and zebrafish (58–60), and could prove beneficial for generating germ-free cnidarians.

One benefit to maintaining a stable but low-diversity microbiome is that we can begin to explore the potential for destabilization of the function of the symbiosis. In our study, we observed that less diverse individuals had reduced fitness in comparison to the controls. However, fitness metrics only started to rebound once the microbiome stabilized in the third week of microbiome recovery. These data support predictions described within the Anna Karenina Principle for animal microbiomes. The Anna Karenina Principle predicts that disturbance events, such as antibiotic exposure, will have a stochastic effect on the animal microbiome leading to microbiome dispersion that will negatively impact host function (61). Thus, microbiome dispersion, rather than alpha diversity alone, plays a key role in defining host health and metabolism (61). While it was beyond the scope of our study to explicitly test the Anna Karenina hypothesis, future experiments building from our work can leverage gnotobiotic Aiptasia to better explore dysbiosis of cnidarian systems.

## Does Aiptasia microbiome have an alternative stable state following antibiotic exposure?

Our antibiotic treatments of Aiptasia resulted in an alternative stable state of the microbiome. The experimental approach transformed a flexible, dynamic microbial community into a microbiome that was narrow but more consistent across the population. For example, samples treated with antibiotics had a reduced population of ASVs belonging to Alteromonadaceae, Rhodobacteraceae, and Flavobacteriaceae families. These bacterial groups are commonly associated with the mucus microbiome of cnidarians, including a major component of the Aiptasia core microbiome (10, 62). Furthermore, species belonging to the Alteromonadaceae likely also play an important role in sulfur cycling by degrading DMSP (63). We also were able to remove members of the marine pathogen family, Vibrionaceae, suggesting that antibiotic treatments may help to restrict the growth of these potentially pathogenic microorganisms within the Aiptasia microbiome. The long-term removal of these bacteria species is promising for performing manipulative experiments to test the function of these species within the metaorganisms. Conversely, some species of Halomonas, Pseudoalteromonas, Rhodobacteraceae, and Winogradskyella were either resistant to the antibiotics and were competitive dominates (e.g., ASV9 and ASV17) or opportunistically colonized or proliferated within the host during recovery (e.g., ASV113 and ASV150). The role and metabolisms of these ASVs within Aiptasia are currently unknown and it is unclear if they are beneficial to the host or cause disease. Consequently, they remain targets for cultivation and cultivation-free analysis (e.g., metagenomics and metatranscriptomics) to determine their putative role within the Aiptasia host.

The establishment of alternate stable states during microbiome recovery from antibiotic treatments is well-studied in vertebrate systems (64–66). Perhaps the best-studied examples are from the human gut, where antibiotic treatments can lead to alternative stable states in microbiome populations, some of which are healthy and others diseased (67, 68). Furthermore, the recovery of the microbiome is often variable,

incomplete (64–66), and can lead to the establishment of antibiotic-resistant microbial species, which can make the host more susceptible to invasions by pathogens or subsequent exposure to stress (67, 68). In our study, we observed a consistent and stable recovery of the Aiptasia microbiome. The observed stability in microbiome recovery could be a result of the limited introduction of outside microorganisms by feeding the anemones with antibiotic-treated brine shrimp and holding them in sterilized artificial seawater, conditions that likely do not mimic recovery dynamics of similarly treated mammalian systems (67, 68). Alternatively, the observed stability may indicate that the animal host has control over the proliferation, or lack thereof, of some microbial lineages after disturbance with the antibiotics. Further work should aim to test different genetic backgrounds of the animal host to explore if genotypic diversity may influence the microbiome recovery trajectory in this emergent model system.

Determining factors that govern the susceptibility or resilience of cnidarians following microbiome manipulations is timely given the growing trend in studies looking to challenge, narrow, or expand the associated microbial community (e.g., heat stress, antibiotic exposure, and probiotics) (27, 33, 69). In our study, we showed that the Aiptasia system represents a powerful model for studying cnidarian resilience with shifting microbial diversity. Despite previous reports, it is possible to generate gnotobiotic individuals with a narrow and defined microbial community, without detrimental consequences to host health. However, it is unclear how the reduction of the microbiome will impact metaorganism response to subsequent disturbance events. Further experiments leveraging multiple exposures to disturbance are needed to quantify the protective role of the host microbiome to stress. Moreover, it is essential to combine these studies with multi-omics approaches to determine mechanisms leading to microbiome disturbance and recovery following antibiotic exposure. Finally, identifying the role of the complexity of the microbiome in cnidarian response to stress remains an essential priority for future studies.

## MATERIALS AND METHODS

### Animal husbandry and experimental overview

We conducted a 76-day-long experiment using *E. diaphana* individuals, originally sourced from Hawaii (H2), to quantify the impacts of two different antibiotic solutions on the Aiptasia microbiome. The experiment was divided into three stages: priming (days 0–33), treatment (days 33–55), and recovery (days 55–76) (Fig. 1). The priming stage enables the initial depletion of the Aiptasia microbiome (43), while exposure to each of the antibiotic solutions during the treatment phase helps to reduce the cnidarian microbial load (33, 35, 42, 47). Throughout the experiment, anemones were reared in six-well plates at 25°C and 12 h light:12 h dark cycle using $44 \pm 15$ µmol photon/m$^2$/s light levels, in either artificial seawater (ASW; control group; Coral Pro Red Sea Salt Salinity 35 ppt) or 0.2 µm filtered artificial seawater (FASW; treatment groups). We chose to rear treatment group anemones in FASW before, during, and after ABS exposure to limit the amount of new bacteria introduced to the anemones during the experiment. This procedure follows previous studies which show that exposing anemones to FASW alone helps to reduce microbial load prior to treatment with antibiotics (33, 43). Anemones were either fed with freshly hatched non-treated (control group) or microbially depleted (treatment group) *A. nauplii* (33). To prevent the build-up of biofilms, water was exchanged from the welled plate after at least 4 h after each feeding and anemones were transferred to fresh sterile plates once per week throughout the experiment.

Following the 33-day priming stage, anemones held in FASW were randomly divided into two groups and treated with either antibiotic solution 1 (ABS1; 50 µg/mL of rifampicin, chloramphenicol, nalidixic acid, and carbenicillin in FASW [33]) or antibiotic solution 2 [ABS2; 50 µg/mL of rifampicin, streptomycin, neomycin, and penicillin in FASW (42)]. In preparation for treatment, mucus from individual anemones was removed via

pipetting, after which the anemone was transferred to a sterile six-well plate. Anemones were then treated with freshly made ABS1, ABS2, or ASW (control) solutions. During the 21-day treatment stage, treatment solutions were changed daily at the start of the dark cycle to avoid photodegradation of the rifampicin.

To determine how long treated anemones remained microbially depleted, we monitored the recovery of the Aiptasia microbiome for 21 days. Treated anemones were held in FASW. During the first 2 weeks of the recovery phase, the anemones were not fed to avoid introducing new microbial species to the previously depleted anemones. After which, they were fed microbially depleted *A. nauplii* twice per week.

## Anemone sampling

To quantify the impact of ABS treatment on the anemone microbiome, we sampled replicate anemones on days 0 (baseline; $n = 3$), 33 (priming; $n = 5$), 55 (treatment; $n = 5$), 61, and 76 (recovery, $n = 5$). To avoid sampling the microbiome associated with the ingestion and digestion of the *A. nauplii*, all anemones were sampled at least four days following the previous feeding day (Fig. 1).

Prior to sampling, anemones treated with ABS solutions were transferred to FASW for 24 h. Individuals were then homogenized in 250 µL of FASW using a pestle motor. An additional 300 µL of FASW was added to each sample to bring the total volume of anemone homogenate to 550 µL. The resulting homogenate was mixed and separated into individual aliquots in preparation for quantifying bacterial load (50 µL), determining algal symbiont densities and protein content (180 µL with 20 µL of 0.1% sodium dodecyl sulfate [SDS]), extracting DNA (250 µL). The homogenate sample prepared for DNA extraction was transferred to a bead beating tube (1:2 mixture lysing matrix B:D MP Biomedicals) containing 500 µL DNA/RNA shield (Zymo Research) and subsequently cells were lysed using a Fastprep 24-5G (MP-Biomedicals; two cycles of 8 m/s for 60 s with a 5-min pause between cycles). All aliquots of the original homogenate, except that used to determine microbial load, were held at −80°C until further analysis. Finally, at each time point, we also sampled 1 L of FASW by collecting filtrate on a 0.22-µm filter cartridge (Millipore-Sigma Cat # SVGP01050) to quantify the composition of the microbial communities within the FASW. After filtration, the cartridge ends were sealed and stored at −80°C until DNA extraction.

## Bacterial load

We quantified the bacterial load from individual anemones using a CFU assay. Briefly, 50 µL of the anemone's homogenate was serially diluted 10-, 100-, 1,000-, and 10,000-fold and 50 µL of each dilution was plated on marine agar plates (Difco Marine Broth 2216 with agar). Plates were incubated at 28°C for at least 24 h and up to two days, after which colonies were counted. We compared CFU counts between treatment groups at each sampling time point independently using a Kruskall-Wallis nonparametric test to determine the effect of treatment (ABS1, ABS2, and ASW) on the number of CFUs/mL.

## Microbiome data collection

In preparation for DNA extraction, the lysed anemone homogenate was defrosted on ice. Filter cartridges were defrosted on ice and DNA/RNA shield was added directly to the filter column at the start of defrosting, then drained into the bead-beating tube. Filters were aseptically removed from the column using pre-sterilized surgical tools (e.g., razor blades and pipe cutters) within a biological safety cabinet and sliced into strips before being placed in the bead-beating tube and homogenized as above. Zymobiomics DNA miniprep kit (Zymo Research) was used to extract DNA from all samples following the manufacturer's protocol with the following modification: DNA was eluted from the binding column in two separate, 1 min incubations using 50 µL of nuclease-free water to result in the final volume of 100 µL. Resulting nucleic acid concentrations were measured

using the Qubit dsDNA (double-stranded DNA) broad range assay kit (Invitrogen) on a Qubit 4 Fluorometer (Invitrogen).

To generate the sequencing library targeting the V4 region of 16S rRNA gene, DNA extracts were amplified in duplicate reactions using the barcoded primer set 515FY, 806RB from the Earth Microbiome Project (EMP) (70, 71). We chose to use the V4 region of the 16sRNA gene for two reasons: first, the EMP primers have the highest fidelity in recovering taxonomic identifications from prokaryote taxa from marine systems (72) and second, by using the EMP primers, we can compare our results to past studies aiming to manipulate or knockdown the microbiome of stony corals (e.g., references (35, 47). Briefly, PCR reactions were composed of 2–4 µL of DNA template, 1.25 µL of each primer, 12.5 µL of Q5 High Fidelity 2X Master Mix (NEB), and 7–9 µL of nuclease-free water for a final volume of 25 µL. The amount of DNA template added to each reaction was dependent on the amount of template needed to achieve amplification. PCR reactions were carried out using a Bio-Rad C1000 thermal cycler (Bio-Rad Laboratories) with the following gradient: an initial denaturation step at 98°C for two minutes followed by 30–38 cycles of denaturation at 98°C for 20 s, 55°C for 15 s, 72°C for 3 min, and a final extension of 72°C for 5 min. All reactions were held at 4°C. Duplicate PCR products that successfully amplified were pooled. About 24 µL of the pooled reactions was visualized on a 1% agarose/TAE gel. Using a Quick-Load Purple 100 bp DNA Ladder (NEB) standard, the band corresponding to the expected size of the V4 region of the 16S rRNA gene was excised from the gel and extracted using the Monarch DNA Gel Extraction Kit (NEB). Purified PCR products were pooled to equal molar ratio (2 ng/µL) and submitted to the University of California, Davis, Genome Center for sequencing using 150 bp paired-end MiSeq 300 platform (Illumina) (9).

## Microbiome analysis

Demultiplexed fastq files were processed in R Studio (v.1.1.463) using the DADA2 pipeline (v.1.24.0) (73) and filtered using the FilterandTrim function using default parameters with the following exceptions: TruncLen = c(150,150), maxEE = c (2,2), and multithread = True. Sequences were subsequently denoised, merged and chimeras were removed, bringing the total number of sequences in the data set from 10,796,554 to 9,768,428. Samples with less than 20,000 reads or greater than 250,000 reads were removed from the data set. 3,012 ASVs were identified in the data set and taxonomy was assigned using the Genome Taxonomy Database R.202 (74). Assigned Taxa, ASVs, and sample data were merged into a single phyloseq object for subsequent analysis (75). After which, ASVs with taxonomic strings matching chloroplasts, mitochondria, and eukaryotic identifications were removed from the data set resulting in 2,800 ASVs. A multiple sequence alignment was run using the AlignSeq function from the DECIPHER package (76), and a neighbor-joining tree was created and fitted for maximum likelihood using a GTR model with the phangorn package (77). The resulting tree was added to the phyloseq object. In order to remove contaminating sequences from our phyloseq object, we used the decontam package to help identify ASVs that are not part of the Aiptasia microbiome using two steps (78). First, using the isContaminant function, we identified 28 ASVs as kit contaminants by applying a prevalence threshold of 0.1 to the extraction blanks. Additionally, we removed 2,280 ASVs with only one occurrence in the data set. Because we also wanted to remove ASVs that were primarily within the filtered ASW, we then performed a second decontamination step that leveraged the seawater blank samples to filter out an additional 80 ASVs based on the same prevalence criteria. After removing the contaminating sequences, we removed all extraction and seawater blanks from the data set which resulted in a phyloseq object of 82 samples composed of 412 ASVs. The full phyloseq object was used for microbial diversity analysis described below.

First, to determine the impact of antibiotics on bacterial alpha diversity we used phyloseq to calculate the observed number of ASVs per sample using the entire 412 ASV data matrix. We then calculated a two-way ANOVA to quantify shifts in the total number

of ASVs across sampling time points and treatment groups. Differences in treatment levels were determined using a Tukey's HSD test.

Next, we chose to quantify shifts in sample beta-diversity of the Aiptasia core microbiome. To calculate the core, we first filtered the phyloseq object to only include samples held in ASW (control) and ASVs with non-zero counts. We then determined the core microbiome using the *core* function within the microbiome package (v1.22.0) (79) by retaining ASVs based on prevalence (i.e., ASVs present >90% of control samples). After normalizing ASV counts to relative abundance within the original phyloseq object, we filtered the ASVs to include only the core community. The resulting phyloseq object was subsequently used to calculate a Bray-Curtis distance matrix that was used to inform a PCoA. Differences between sample groups were tested using a PERMANOVA using the adonis2 function within vegan (v.2.6.4) (80). We also calculated and compared the distance to the group centroid using the betadisper function in order to determine within-group variance of the microbiome.

Finally, to determine which taxa are differentially enriched directly after treatment (55 days) and during recovery (61 and 76 days), we used the Aldex2 package (v.1.28.1) (44) to compare variation in the abundance of ASVs between the control and each treatment group at each sampling time point. We chose to use the entire data sets for the differential abundance analysis to ensure we were not missing key taxa that were not part of the core that either significantly increased or decreased in abundance during the experiment. Taxa were determined to be significant if they had an effect size >2 and a $P < 0.05$ for both aldex.ttest and aldex.glm at any of the three time points for either the control versus ABS1 and control versus ABS2 comparisons. After determining which taxa were differentially abundant, we plotted relative abundances across treatment groups and time points (55, 61, and 76 days) and classified individual taxa into groups based on their shift in relative abundance directly after treatment and during recovery.

## Aiptasia physiology

To quantify the impacts of the antibiotics on host metaorganism physiology, we monitored shifts in total protein, algal densities, and asexual reproduction rates throughout the experiment. To assess shifts in biomass, anemone homogenate was defrosted and 20 µL was used in a Pierce BCA protein assay to quantify total protein in triplicate according to the manufacturer's instructions (Thermo Scientific). To quantify algal cell density, we defrosted 180 µL of the anemone homogenate reserved for algal cell counts and diluted the homogenate 1:20 in FASW containing 0.01% SDS. Animal tissue cells in the diluted surrey were sheared with a 25G needle. Algal cell counts were obtained from 100 µL of each sample using flow cytometry (Bio-Rad ZE5 Cell Analyzer) using the following parameters: 405 nm laser forward scatter, 448 nm laser side scatter, and Chlorophyll fluorescence with a 448 nm laser excitation and detection with a 692/80 bandpass filter. The plate was agitated every five samples and ran at a flow rate of 1.5 µL/s and resulting cell counts were normalized to total protein content (81). To quantify the rate of asexual reproduction, we held a separate set of replicate ($n = 6$) anemones in six-well plates and treated them as described above for the experimental anemones. Throughout the experiment, we counted the number of produced pedal lacerates on a weekly basis. Counted pedal lacerates were removed from each well following counts. All metrics were compared across treatment groups and sampling time points after being checked for normality using a Shapiro-Wilk test. Non-normal data sets (CFU counts and pedal lacerates) used a Kruskal-Wallis test with Dunn's post hoc comparison, while normally distributed parameters (protein content and log-adjusted symbiont density) used an ANOVA with Tukey's honest significant difference using the stats package in R.

## ACKNOWLEDGMENTS

The authors would like to thank the UC Merced Stem Cell Instrumentation Foundry and Dr. David Gravano for assistance in generating flow cytometry data. The sequencing was carried out at the UC Davis Genome Center DNA Technologies and Expression Analysis Core, supported by NIH Shared Instrumentation Grant 1S10OD010786-01. The authors thank Dr. Anya Brown for fruitful discussions related to the manuscript.

Research was supported in part by the DoD Research and Education Program for HBCU/MSI Instrumentation Grant W911NF1910529. Funding from the National Science Foundation Grant to M.S. (NSF BRC-Bio #2217769) provided financial support for this work.

## AUTHOR AFFILIATION

[1]Department of Molecular Cell Biology, University of California, Merced, California, USA

## AUTHOR ORCIDs

Sophie MacVittie http://orcid.org/0000-0002-4925-009X
Maggie Sogin http://orcid.org/0000-0001-7533-3705

## FUNDING

| Funder | Grant(s) | Author(s) |
| --- | --- | --- |
| National Science Foundation (NSF) | 2217769 | Maggie Sogin |

## DATA AVAILABILITY

Raw sequences are available at the National Center for Biotechnology Information's Sequence Read Archive under PRJNA1048100. Physiological data presented in this paper can be found at https://doi.org/10.5281/zenodo.10257452. Code is available at https://github.com/smacvittie/aiptasia_abs_recovery.

## ADDITIONAL FILES

The following material is available online.

### Supplemental Material

**Supplemental Information (mSystems01342-23-s0001.docx).** Legends for supplemental tables.
**Supplemental tables (mSystems01342-23-s0002.xlsx).** Tables S1 to S4.

### Open Peer Review

**PEER REVIEW HISTORY (review-history.pdf).** An accounting of the reviewer comments and feedback.

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
