## [Reviewer comments · mSystems]

Microbiome depletion and recovery in the sea anemone, *Exaiptasia diaphana*, following antibiotic exposure

Sophia MacVittie, Saam Doroodian, Aaron Alberto, and E. Sogin

Corresponding Author(s): E. Sogin, Univ. California Merced

Review Timeline:

Submission Date:	December 11, 2023
Editorial Decision:	March 3, 2024
Revision Received:	April 15, 2024
Accepted:	April 19, 2024

Editor: Laetitia Wilkins

Reviewer(s): Disclosure of reviewer identity is with reference to reviewer comments included in decision letter(s). The following individuals involved in review of your submission have agreed to reveal their identity: Emily G Aguirre (Reviewer #1)

Transaction Report:

DOI: <https://doi.org/10.1128/msystems.01342-23>

Re: mSystems01342-23 (**Microbiome depletion and recovery in the sea anemone, *Exaiptasia diaphana*, following antibiotic exposure**)

Dear Dr. E. Maggie Sogin:

Revision Guidelines

Sincerely,
Laetitia Wilkins
Editor
mSystems

Reviewer #1 (Comments for the Author):

Microbiome studies in general, are common but to my knowledge, this is among a first handful of studies exploring the effects of antibiotics on the fitness of Aiptasia. Therefore, this research is original and addresses a question of broad interest to the cnidarian community, "Can we use antibiotics to create more manageable alternative coral model system using Aiptasia".

INTRODUCTION

(L77-L82): I would suggest rewording here, as Symbiodiniaceae are not "thought" to play a fundamental role. By now, we know they play a role. This is a fact not up for debate given the overwhelming evidence and I wouldn't place it in speculative territory.

(L100): Hydra is not an anemone; it is a cnidarian but not anemone. Please revisit references.

(L120-122): I'd suggest from putting a number on how many different types of microbes Aiptasia hosts, since the numbers can vary widely. Please rephrase this.

(L138): Would you be able to briefly mention the fitness metrics used?

RESULTS

Comment (my thoughts, not suggestion for any changes) Here, I would have liked to know more about the effect of antibiotics on the animal itself. Probably an elaboration on what it means to have lower protein and lower pedal lacerate formation in antibiotics and if this, in turn, also disturbs microbiome recruitment.

Fig. 2a: 61d a and ab are similar in ASVs. Do you know why this would be?

(L336): Please be consistent with how p-values are presented.

(L386): Please be consistent on how statistical measures are presented here and throughout the text. Consult a statistical handbook on how to present the results.

(L432): Please detail in discussion why these taxa were deemed opportunistic. Have these taxa showed up in other Aiptasia microbial surveys? Is there a possibility they may be part of the rare microbiome?

DISCUSSION

(L475-477): Please elaborate further on the comment about Hydra. Maybe add in another sentence on how this supports this finding in Aiptasia. As of now, it seems rather awkwardly placed in the middle of the discussion.

(L484): Hydra is not an anemone.

(L524-L526): Here you could probably quote some of Monica Medina's group recent work on the upside-down jellyfish, *Cassiopea xamachana* and Patricia E. Thome's group.

(L530-L533 and L567-L570): Most likely explanations. I would have liked to see a dive into what is known about antimicrobial resistance in the "resilient" bacteria that persisted. Are you concerned about accidentally creating superbugs in the quest for a microbial model for Aiptasia?

(L575-L577): Please provide more sentences and references qualifying this statement that starts with "Furthermore..."

MATERIALS AND METHODS

Fig.1: I'd suggest making these into two separate figures and providing detailed explanation there. It's a bit too much for one figure.

(L594-L622): Why was the same type of media (0.2 um FASW) not used for everyone Media can influence microbial composition.

(L655): CFU counts are only accurate if the media agrees with the bacteria. Many bacteria in anemones may be missed in Marine Agar plates, like Difco, outcompeted by fast growers. IMO some anemone-associated bacteria also start appearing on plates up to 5-7 days after inoculation. Please address the issue of uncultured microbes/compatibility with the media used in your discussion.

(L664-L666): Please elaborate on this protocol to assure sterility during this process. IMO this technique can result in some contamination from surgical tools.

(L674-L675): This primer set misses many bacterial species or will preferentially amplify chloroplasts and mitochondria in Aiptasia, possibly skewing detection of certain types of bacterial species. Could you provide info on why this primer set was used, other than everyone else uses it...

(L694-L695): Would you elaborate as to why `truncLen=c(150,150)` was employed and not `truncLen=c(240,160)`, as recommended by the DADA2 tutorial? This can affect ID of your microbes.

(L702-705): Would you elaborate as to why an external package was used to build the tree instead of the native function in Phyloseq? Sometimes imports into Phyloseq object can cause artifacts...

(L697-698): How was the data normalized? That's fine but only if there's an alternate way to normalize the data. Working with a range of 20k to 250k is significant. Would you elaborate as to why normalization was not conducted? This may be OK if arriving to the same biological conclusion, regardless of normalization.

Reviewer #2 (Comments for the Author):

The authors conducted a comparative analysis of two antibiotic treatments and explored the phenotypic and microbial alterations in *Exaipasia disphanna* during the recovery phase. They observed a reduction in microbial diversity during this phase, yet the community successfully transitioned to a new stable state. However, the depth of the microbiome analysis for Aiptasia, as a

model organism for marine cnidarians, seems insufficient. To unveil the response patterns of Aiptasia's microbiome to antibiotic treatment, a more comprehensive understanding can be achieved through the integration of metagenomic and metabolomic approaches. This would enable the identification of alterations in functional genes and metabolic products associated with Aiptasia's microbial community. Such an approach could provide direct evidence for uncovering the adaptive mechanisms of Aiptasia's symbiotic functional body in response to antibiotic stress.

Materials and Methods

Lines 147-149: The selection of ABS1 (carbenicillin, chloramphenicol, nalidixic acid, and rifampicin) and ABS2 (neomycin, penicillin, rifampicin, streptomycin) as antibiotics for treatment needs clarification. What do these antibiotics represent, and can their concentrations in natural water reach the levels used in the experiment? These considerations are pivotal for assessing the accuracy of the experimental methods.

Lines 146-150: The durations of artificial seawater treatment, antibiotic exposure, and the recovery phase were 33 days, 22 days, and 21 days, respectively. What is the rationale behind choosing different durations for these experimental phases?

Discussion

Lines 528-539: Exploring whether maintaining low microbial diversity in Aiptasia over the long term is beneficial or detrimental to the health of the symbiotic functional body would benefit from a more in-depth discussion, possibly integrating an open microbial system with the "Anna Karenina" principle.

-The author leaves several questions unanswered in the discussion, such as "However, it is unclear how the reduction of the microbiome will impact metaorganism response to subsequent disturbance events" (lines 588-589) and "The role and metabolisms of these ASVs within Aiptasia are currently unknown and remain targets for cultivation and cultivation-free analysis" (lines 559-561). Addressing these questions could be achieved with additional experimental work.

-The findings appear to echo patterns observed in previous research on cnidarians. Utilizing a multi-omics approach for the model organism Aiptasia could provide a more comprehensive understanding of the response and adaptation patterns of the symbiotic functional body to antibiotic treatment.

Comments to the authors

INTRODUCTION

(L77-L82): I would suggest rewording here, as Symbiodiniaceae are not “thought” to play a fundamental role. No, we know they play a role. This is a fact not up for debate given the overwhelming evidence and I wouldn’t place it in speculative territory.

(L100): *Hydra* is not an anemone; it is a cnidarian but not anemone. Please revisit references.

(L120-122): I’d suggest from putting a number on how many different types of microbes *Aiptasia* hosts, since the numbers can vary widely. Please rephrase this.

(L138): Would you be able to briefly mention the fitness metrics used?

RESULTS

Comment (my thoughts, not suggestion for any changes) → Here, I would have liked to know more about the effect of antibiotics on the animal itself. Probably an elaboration on what it means to have lower protein and lower pedal lacerate formation in antibiotics and if this, in turn, also disturbs microbiome recruitment.

Fig. 2a: 61d a and ab are similar in ASVs. Do you know why this would be?

(L336): Please be consistent with how p-values are presented.

(L386): Please be consistent on how statistical measures are presented here and throughout the text. Consult a statistical handbook on how to present the results.

(L432): Please detail in discussion why these taxa were deemed opportunistic. Have these taxa showed up in other *Aiptasia* microbial surveys? Is there a possibility they may be part of the rare microbiome?

DISCUSSION

(L475-477): Please elaborate further on the comment about *Hydra*. Maybe add in another sentence on how this supports this finding in *Aiptasia*. As of now, it seems rather awkwardly placed in the middle of the discussion.

(L484): *Hydra* is not an anemone.

(L524-L526): Here you could probably quote some of Monica Medina’s group recent work on the upside-down jellyfish, *Cassiopea xamachana* and Patricia E. Thome’s group.

(L530-L533 and L567-L570): Most likely explanations. I would have liked to see a dive into what is known about antimicrobial resistance in the “resilient” bacteria that persisted. Are you concerned about accidentally creating superbugs in the quest for a microbial model for *Aiptasia*?

(L575-L577): Please provide more sentences and references qualifying this statement that starts with "Furthermore..."

MATERIALS AND METHODS

Fig.1: I'd suggest making these into two separate figures and providing detailed explanation there. It's a bit too much for one figure.

(L594-L622): Why was the same type of media (0.2 um FASW) not used for everyone Media can influence microbial composition.

(L655): CFU counts are only accurate if the media agrees with the bacteria. Many bacteria in anemones may be missed in Marine Agar plates, like Difco, outcompeted by fast growers. IMO some anemone-associated bacteria also start appearing on plates up to 5-7 days after inoculation. Please address the issue of uncultured microbes/compatibility with the media used in your discussion.

(L664-L666): Please elaborate on this protocol to assure sterility during this process. IMO this technique can result in some contamination from surgical tools.

(L674-L675): This primer set misses many bacterial species or will preferentially amplify chloroplasts and mitochondria in Aiptasia, possibly skewing detection of certain types of bacterial species. Could you provide info on why this primer set was used, other than everyone else uses it...

(L694-L695): Would you elaborate as to why `truncLen=c(150,150)` was employed and not `truncLen=c(240,160)`, as recommended by the DADA2 tutorial? This can affect ID of your microbes.

(L702-705): Would you elaborate as to why an external package was used to build the tree instead of the native function in Phyloseq? Sometimes imports into Phyloseq object can cause artifacts...

(L697-698): How was the data normalized? That's fine but only if there's an alternate way to normalize the data. Working with a range of 20k to 250k is significant. Would you elaborate as to why normalization was not conducted? This may be OK if arriving to the same biological conclusion, regardless of normalization.

Response to reviewers

Reviewer #1 (Comments for the Author):

Microbiome studies in general, are common but to my knowledge, this is among a first handful of studies exploring the effects of antibiotics on the fitness of *Aiptasia*. Therefore, this research is original and addresses a question of broad interest to the cnidarian community, "Can we use antibiotics to create more manageable alternative coral model system using *Aiptasia*".

We thank the reviewer for their kind comments and agree that our study represents an important step in developing alternative models for studying cnidarian-microbiome interactions.

INTRODUCTION

1. (L77-L82): I would suggest rewording here, as Symbiodiniaceae are not "thought" to play a fundamental role. By now, we know they play a role. This is a fact not up for debate given the overwhelming evidence and I wouldn't place it in speculative territory.

We agree with the reviewer that there is overwhelming evidence that Symbiodiniaceae play a critical role in supporting the metabolism of many cnidarian species. We have reworded L77-L82 to state:

L77-L82: "Some of these microbial taxa play a fundamental role in supporting host health and metabolism. The most famous of which is the relationship between reef-building corals and their dinoflagellate, microalgal symbionts in the family Symbiodiniaceae. Overwhelming evidence demonstrates that the microalgal partner supports coral growth and survival through the transfer of sugars to the animal host (Muscatine et al. 1984, Davy et al. 1996)."

We have also added the following references in support of this statement:

- (1) Muscatine L, Falkowski PG, Porter JW, Dubinsky Z. Fate of photosynthetic fixed carbon in light-and shade-adapted colonies of the symbiotic coral *Stylophora pistillata*. *Proceedings of the Royal Society of London. Series B. Biological Sciences.* 1984 Dec 31;222(1227):181-202.
- (2) Davy SK, Lucas IA, Turner JR. Carbon budgets in temperate anthozoan-dinoflagellate symbioses. *Marine Biology.* 1996 Oct;126:773-83.

2. (L100): Hydra is not an anemone; it is a cnidarian but not anemone. Please revisit references.

We thank the reviewer for catching our mistake and have made the following edits:

L100: "For example, the depletion of Hydra's microbiome revealed that microbial taxa were (i) critical in defending the animal against fungal pathogens..."

L502: "In the freshwater cnidarian, Hydra, ..."

3. (L120-122): I'd suggest from putting a number on how many different types of microbes Aiptasia hosts, since the numbers can vary widely. Please rephrase this.

We appreciate the reviewer's suggestion to clarify the exact number of microbes associated with Aiptasia and have made the following change in the text to indicate that the Aiptasia microbiome is quite complex:

L121: "Aiptasia also hosts a complex core microbiome, likely consisting of 24-44 bacterial species with likely 100s to 1000s of accessory members (as defined by amplicon sequence variants or operational taxonomic units) that live in the anemone's tissue and mucus layers (10, 32, 38, 39)."

One issue in trying to determine the exact number of microbial species within a host microbiome is that the methods used (DNA extraction, library generation, and bioinformatics) can vary considerably across studies. Furthermore, given biases in PCR implications and challenges with conducting metagenomic studies on hosts that associate with complex communities may restrict our ability to definitively state the exact number or range in total microbial units a specific host associates with.

4. (L138): Would you be able to briefly mention the fitness metrics used?

We agree that briefly mentioning the fitness metrics used will help guide the reader. We have made the following change to the manuscript:

L139: "...which resulted in fitness declines, as measured in total biomass, algal cell density, and asexual reproduction metrics."

RESULTS

5. Comment (my thoughts, not suggestion for any changes) ◇ Here, I would have liked to know more about the effect of antibiotics on the animal itself. Probably an elaboration on what it means to have lower protein and lower pedal lacerate formation in antibiotics and if this, in turn, also disturbs microbiome recruitment.

6. Fig. 2a: 61d a and ab are similar in ASVs. Do you know why this would be?

We agree with the reviewer that this is an interesting finding to speculate on. In Figure 3a we showed the total number of ASVs per treatment per sampling day per anemone. Importantly, our results show that the diversity of the microbiome is not impacted while the anemones are held in ABS solutions, despite the clear decline in bacterial load. Rather, the total number of ASVs only declined during the recovery phase. For ABS1, we see that the microbial diversity is significantly lower than the control group on day 61 (or 1 week post ABS exposure). This was largely driven by the domination of the community by a member of the Stappiaceae family (See

Fig 4). For ABS2, we only observed a significant decline in the number of ASVs on day 76 (or three weeks post ABS exposure). The differences between ABS1 and ABS2 could be due to different types of antibiotics that differentially remove key bacterial groups, which would impact the recovery dynamics of the microbiome as the bacterial population rebounded.

At this stage we can only speculate why microbiome diversity declines in Aiptasia following ABS exposure during the recovery stage as the bacterial population rebounds, which we outline in our discussion in lines 529-534 which reads:

“Our work indicates that the recovery period following antibiotic exposure is critical for the establishment of a less complex microbiome model for Aiptasia. Recovery allows for the stabilization of the microbial communities and is likely impacted by the members of the microbiome that are differentially removed versus retained. Interestingly, not all cnidarians or marine invertebrates can maintain a low-diversity microbiome after antibiotic exposure...”

7. (L336): Please be consistent with how p-values are presented.

We thank the reviewer for catching our mistake in how we report p-values. We have adjusted this throughout the text and p-values are now always reported as “Test Used $p = X$ ” or “ $p < 0.X$ ”.

8. (L386): Please be consistent on how statistical measures are presented here and throughout the text. Consult a statistical handbook on how to present the results.

We have adjusted to the text to be consistent with how statical measures are presented:

“(BH adjusted Welch’s t-test $p < 0.05$)” or “BH adjusted $p < 0.05$ ” and is consistent throughout the legend and text.

9. (L432): Please detail in discussion why these taxa were deemed opportunistic. Have these taxa showed up in other Aiptasia microbial surveys? Is there a possibility they may be part of the rare microbiome?

In our study, we chose to define opportunistic taxa as taxa that were able to grow and proliferate once other members of the microbiome were removed due to antibiotic exposure. This only accounted for two ASVs in our study, ASV 150 and ASV 113 and only occurred in the ABS2 treatment. While possible that these ASVs were part of the rare biosphere, we did not observe either of them in any other samples except those taken at the final recovery time point following exposure to ABS2 (Fig. 6). To the best of our knowledge, these ASVs have not been observed in other Aiptasia microbial surveys (however certainly other Aiptasia microbiome studies have found close relatives).

Based on the reviewer comments, we modified Lines 594-601 which now read:

“Conversely, some species of Halomonas, Pseudoalteromonas, Rhodobacteraceae, and Winogradskyella were either resistant to the antibiotics and were competitive dominantes

(e.g., ASV9, ASV17) or opportunistically colonized or proliferated within the host during recovery (e.g., ASV113, 150). The role and metabolisms of these ASVs within Aiptasia are currently unknown and it is unclear if they are beneficial to the host or cause disease. Consequently, they remain targets for cultivation and cultivation-free analysis to determine their putative role.”

DISCUSSION

10. (L475-477): Please elaborate further on the comment about Hydra. Maybe add in another sentence on how this supports this finding in Aiptasia. As of now, it seems rather awkwardly placed in the middle of the discussion.

We agree with the reviewer that this statement was not fully explored in the context of our results. We aimed to use the Hydra example for two purposes, first to show that antibiotics reduce feeding in sea anemones and hydra and second to showcase that the antibiotic solutions may pose more serious threats to Aiptasia metabolism as they are unable to feed even when transferred to fresh media.

We have expanded Lines 491-497 to now read:

“Our results are consistent with similar observations from Hydra, namely hydra polyps held in antibiotic solutions were unable to feed on Artemia, however regained their feeding phenotype once transferred to fresh media (46). We observe a similar behavior in Aiptasia, however antibiotic exposure in our anemones may have more serious phenotypic consequences than their freshwater relatives as Aiptasia seem to halt or significantly reduce their feeding behavior even when held in antibiotic-free media.”

11. (L484): Hydra is not an anemone.

We thank the reviewer for catching our mistake and have made the appropriate adjustment.

12. (L524-L526): Here you could probably quote some of Monica Medina's group recent work on the upside-down jellyfish, *Cassiopea xamachana* and Patricia E. Thome's group.

We thank the reviewer for directing us to find related work from Dr. Medina and Dr. Thome’s groups. While we did not specifically find studies to support our discussion point surrounding the need for longitudinal work showing recovery dynamics of cnidarian microbiomes, we did find an interesting study that could explain some of the observed variation in cnidarians around microbiome recovery. We have added Line 542 to the manuscript which now reads:

“Variation in organismal responses across cnidarian groups may be due to variation in the production of antimicrobial compounds within the surface mucus layers (Rivera-Ortega and Thome 2018)”

1. *Rivera-Ortega, J and Thome, PE (2018) Contrasting antibacterial capabilities of the surface mucus layer from three symbiotic cnidarians. Front. Mar. Sci. DOI: 10.3389/fmars.2018.00392/full*

13. (L530-L533 and L567-L570): Most likely explanations. I would have liked to see a dive into what is known about antimicrobial resistance in the "resilient" bacteria that persisted. Are you concerned about accidentally creating superbugs in the quest for a microbial model for *Aiptasia*?

*We thank the reviewer for this comment because it helps to provide additional context for our results. Given that we are working with tag-sequencing data (and not metagenomes), we only looked at susceptible and resistant taxa where we were able to assign a full taxonomic string to develop hypotheses on if the ASVs identified in our *Aiptasia* might have the capacity to resist antibiotics. Of the eight ASVs that both had a full taxonomic string and were part of the core microbiome, four matched taxa that are known to contain antibiotic resistant genes. Two of these four were considered resistant taxa in our analysis (ASV11, ASV16).*

To reflect the suggestion of the reviewer, we added Line 553:

*“Indeed, two of the ASVs that were either not removed during antibiotic exposure or were considered resistant taxa were closely related to bacterial species that were either shown to be resistant to a full suite of antibiotic drugs (e.g., *Marisediminitalea aggregate*, Zhang et al. 2020) or contained antibiotic genes within their genomes (e.g., *Aureliella hegolandensis*, Kallscheuer et al. 2020).”*

*In terms of the reviewer’s question if we are concerned about generating superbugs, we do not have data currently to either support or refute this concept, however studies in the past using antibiotics to try to knockdown the *Aiptasia* microbiome (e.g., Hartman et al. 2022) also observed antibiotic resistant strains of bacteria, including members of the *Vibrio* genus. The presence of antimicrobial compounds found in the surface mucus layer, produced either by the host or the microbes themselves may also help to explain the natural resistance of some host-associated microbes. This phenomenon is found consistently across cnidaria as reviewed in Mariottini et al. 2016, Stabili et al. 2018, Voolstra et al. 2024.*

*Mariottini, G.L.; Grice, I.D. Antimicrobials from Cnidarians. A New Perspective for Anti-Infective Therapy? Mar. Drugs **2016**, *14*, 48. <https://doi.org/10.3390/md14030048>*

*Stabili, L.; Parisi, M.G.; Parrinello, D.; Cammarata, M. Cnidarian Interaction with Microbial Communities: From Aid to Animal’s Health to Rejection Responses. Mar. Drugs **2018**, *16*, 296. <https://doi.org/10.3390/md16090296>*

Voolstra, C.R., Raina, JB., Dörr, M. et al. The coral microbiome in sickness, in health and in a changing world. Nat Rev Microbiol (2024). <https://doi.org/10.1038/s41579-024-01015-3>

14. (L575-L577): Please provide more sentences and references qualifying this statement that starts with "Furthermore..."

We are unsure which line the reviewer is referring to in this case as line 575-577 starts either with "Alternatively" (now Line 615) or "Future work" (now Line 617) and represent ideas that we have generated while analyzing the data presented in the current manuscript.

MATERIALS AND METHODS

15. Fig.1: I'd suggest making these into two separate figures and providing detailed explanation there. It's a bit too much for one figure.

We agree with the reviewer that Figure 1 is complex and decided to break it up into two figures. All other figures are now renamed in the main text file accordingly.

16. (L594-L622): Why was the same type of media (0.2 um FASW) not used for everyone
Media can influence microbial composition.

We agree with the reviewer that the media type can influence the microbial composition and, importantly, the bacterial load of our anemones. Because the goal of our experiment was to reduce and alter host-associated microbiomes, and not quantify just the impacts of antibiotics on host physiology, we implemented protocols from past studies (e.g., Costa et al. 2021; Dugan et al. 2020) to reduce microbial load which included rearing treatment anemones in filtered or sterile ASW and compare them to a control (e.g., normal treatment) group. To better articulate our goals in the text we added line 648 to the manuscript which reads:

"We chose to rear treatment group anemones in FASW before, during and after ABS exposure to limit the amount of new bacteria introduced to the anemones during the experiment. This procedure follows previous studies which show that exposing anemones to FASW alone helps to reduce microbial load prior to treatment with antibiotics (Costa et al. 2021; Dugan et al. 2020)."

1. Costa RM, Cárdenas A, Loussert-Fonta C, Toullec G, Meibom A, Voolstra CR. 2021. Surface topography, bacterial carrying capacity, and the prospect of microbiome manipulation in the sea anemone coral model *Aiptasia*. *Frontiers Microbiology* 12:637834.
2. Dungan A, van Oppen M, Blackall L. 2021. Short-Term Exposure to Sterile Seawater Reduces Bacterial Community Diversity in the Sea Anemone, *Exaiptasia diaphana*. *Frontiers in Marine Science* 7.

17. (L655): CFU counts are only accurate if the media agrees with the bacteria. Many bacteria in anemones may be missed in Marine Agar plates, like Difco, outcompeted by fast growers. IMO some anemone-associated bacteria also start appearing on plates up to 5-7 days after inoculation. Please address the issue of uncultured microbes/compatibility with the media used in your discussion.

We agree with the reviewer CFU counts are a crude metric for assessing bacterial load and determine absolute carrying capacity of a host's or environmental sample's microbiome. However, our goal was not to determine the absolute carrying capacity of the microbiome across each sample, but rather to compare CFU counts between treatments to determine if we were able to reduce the relative microbial load with the addition of antibiotics. In this case, we chose to use a culture approach to quantify microbiome reduction.

We added line 462 in our discussion to reflect the difficulty in culturing all resident microbial taxa using marine agar plates:

"It is likely that our colony forming unit assay, which allowed us to show that bacterial load is reduced in treated anemones, missed key bacterial groups that remain refractory to cultivation on marine agar. However, our results are consistent with previous work in cnidarians..."

18. (L664-L666): Please elaborate on this protocol to assure sterility during this process. IMO this technique can result in some contamination from surgical tools.

We were also concerned that the cutting open of the filter cartridges could result in contamination of the resulting sequencing dataset. To minimize these impacts, all surgical tools used to cut open the filters (e.g., razor blades, pipe cutters) were sterilized using ethanol and then flame sterilized within a Biological Safety Cabinet. Furthermore, we made sure to use autoclaved or pre-sterilized materials (e.g., petri dishes) during the procedure to reduce input from the environment. We'd also like to highlight that the sequencing data from these samples was used to remove ASVs bioinformatically that were associated with the FASW media using the decontam package. As such, if there was contamination during the extraction phase of this experiment, the environmental bacterial associated with sample processing (and not FASW) would not have impacted downstream analyses.

We have modified line 709 to better explain our aseptic approach:

"Filters were aseptically removed from the column using pre-sterilized surgical tools (e.g., razor blades and pipe cutters) within a biological safety cabinet and sliced into strips before being placed in the bead beating tube and homogenized as above."

19. (L674-L675): This primer set misses many bacterial species or will preferentially amplify chloroplasts and mitochondria in Aiptasia, possibly skewing detection of certain types of bacterial species. Could you provide info on why this primer set was used, other than everyone else uses it...

We agree with the reviewer that there is no "perfect" primer set for determining all microbial components of a specific sample, especially in the case of host-associated communities. However, alternative approaches, like metagenomic analyses, will likely miss rare members of highly complex communities or are cost prohibitive for the number of samples used in the current study. We chose to use the V4 primers as recommended from the Earth microbiome project (515FY/806RB) for several reasons.

First, we wanted to use primers typically used to study reef-building corals in order to compare our results to past studies that have applied antibiotic solutions to knockout or manipulate cnidarian microbiomes (e.g., Bent et al. 2021; Connelly et al. 2023).

Second, a recent study from McNicol et al. 2021 conducted an in-silico study to quantify the accuracy of different primer pairs in matching the SSU region recovered from oceanic metagenomes as part of the TARA oceans project. Their findings are important for two reasons: (1) they found that different primer regions have a similar potential to generate taxonomic profiles and (2) both the V4 and V4/V5 primer pairs resulted in the highest consistency of perfect matches to the SSUs generated from the metagenomic libraries.

We'd also like to highlight that we took extensive precautions to reduce the number of chloroplast and mitochondria reads in the sequencing dataset prior to generation of the library. As described on lines 729-733, before generating the sequencing library, each PCR products were run on a 1% agarose gel. We then excised the resulting PCR band at ~400 bp, which corresponds to the expected size of the V4 region with attached primers and barcodes. Only this band was cleaned up and used to generate the library. This procedure has the advantage of excluding primer dimers from the resulting PCR reaction and, critically, the larger bands corresponding to host mitochondria reads and algal chloroplast reads.

As evidence that our gel excision procedure worked, we only needed to bioinformatically remove 212 ASVs of the initial 3012 ASVs that corresponded to either chloroplasts, mitochondria or eukaryotes (Line 748). In total the number of ASVs removed only accounted for 7% of the total reads in the entire dataset.

To clarify our choice in 16S rRNA primers, we added the following statement to the manuscript starting on Line 719:

“To generate the sequencing library targeting the V4 region of 16S rRNA gene, DNA extracts were amplified in duplicate reactions using the barcoded primer set 515FY, 806RB from the Earth Microbiome Project (EMP)(66, 67). We chose to use the V4 region of the 16sRNA gene for two reasons: first, the EMP primers have the highest fidelity in recovering taxonomic identifications from prokaryote taxa from marine systems (McNicol et al. 2021) and second, by using the EMP primers we can compare our results to past studies aiming to manipulate or knockdown the microbiome of stony corals (e.g., Bent et al. 2021; Connelly et al. 2023).”

- 1. McNicol J, Berube PM, Biller SJ, Fuhrman JA. 2021. Evaluating and Improving Small Subunit rRNA PCR Primer Coverage for Bacteria, Archaea, and Eukaryotes Using Metagenomes from Global Ocean Surveys. *mSystems* 6:10.1128/msystems.00565-21.*
- 2. Bent SM, Miller CA, Sharp KH, Hansel CM, Apprill A. 2021. Differential Patterns of Microbiota Recovery in Symbiotic and Aposymbiotic Corals following Antibiotic Disturbance. *mSystems*.*

3. Connelly MT, Snyder G, Palacio-Castro AM, Gillette PR, Baker AC, Traylor-Knowles N. 2023. Antibiotics reduce *Pocillopora* coral-associated bacteria diversity, decrease holobiont oxygen consumption and activate immune gene expression. *Molecular Ecology* 32:4677–4694.

20. (L694-L695): Would you elaborate as to why `truncLen=c(150,150)` was employed and not `truncLen=c(240,160)`, as recommended by the DADA2 tutorial? This can affect ID of your microbes.

We thank the reviewer for asking this question because it allowed us to catch a mistake in our methods section. Namely, our sequencing approach used a MiSeq 300 platform, where we generated paired end reads that were only 150 bp long (not 250 bp). This sequencing approach allowed us to cover the full 291 bp length of the V4 region. Importantly, the DADA2 pipeline is working with PE reads that were 250 bp not the 150 bp of our read set. Therefore, in order to retain reads for our analyses, we needed to change the default values to 150 which allowed us to use the entire read length. Luckily, the quality of our reads was excellent (see figure below), even at the read ends, which allowed us to avoid trimming the read length. On average, our pipeline retained 90% of the sequencing reads.

Figure 1 Above are two quality score plots (left -Forward reads, right - reverse reads) generated during the DADA2 pipeline showing the quality score (y-axis) by bp position of the read (x-axis) for a subset of the datafiles.

21. (L702-705): Would you elaborate as to why an external package was used to build the tree instead of the native function in Phyloseq? Sometimes imports into Phyloseq object can cause artifacts...

To the best of our knowledge, *phyloseq* is unable to construct multisequence alignments that are needed to build phylogenetic trees that we used in our analysis. Therefore, we constructed the tree using a combination of *DECIPHER* package (to construct the multisequence alignment and *phangorn* to build the maximum likelihood tree. We then imported the tree into *phyloseq* to visualize the tree using the `plot_tree` function.

Line 754 reads:

“A multiple sequence alignment was run using the AlignSeq function from the DECIPHER package (71), and a neighbor joining tree was created and fitted for maximum likelihood using a GTR model with the phangorn package (72). The resulting tree was added into the phyloseq object and visualized using the plot_tree function.”

22. (L697-698): How was the data normalized? That's fine but only if there's an alternate way to normalize the data. Working with a range of 20k to 250k is significant. Would you elaborate as to why normalization was not conducted? This may be OK if arriving to the same biological conclusion, regardless of normalization.

We agree with the reviewer that there are many ways to normalize the data. We found data normalization a particular challenge in this type of dataset where we are aiming to compare shifts in relative abundance of microbial taxa that have vastly different read counts, likely due to the fact that we aimed to reduce the microbial load in one treatment group. However, to not “lose” data as recommended by McMurdie and Holmes 2014, we did normalize the resulting read count ASV table to relative abundance within each sample. Our normalization procedure is articulated on line 755 (now line 782) and was conducted prior to beta-diversity and differential abundance analyses.

Line 782:

“After normalizing ASV counts to relative abundance within the original phyloseq object, we filtered the ASVs to only include the core community.”

1. *McMurdie PJ, Holmes S (2014) Waste Not, Want Not: Why Rarefying Microbiome Data Is Inadmissible. PLoS Comput Biol 10(4): e1003531. <https://doi.org/10.1371/journal.pcbi.1003531>*

Reviewer #2 (Comments for the Author):

The authors conducted a comparative analysis of two antibiotic treatments and explored the phenotypic and microbial alterations in *Exaiptasia diaphana* during the recovery phase. They observed a reduction in microbial diversity during this phase, yet the community successfully transitioned to a new stable state. However, the depth of the microbiome analysis for *Aiptasia*, as a model organism for marine cnidarians, seems insufficient. To unveil the response patterns of *Aiptasia*'s microbiome to antibiotic treatment, a more comprehensive understanding can be achieved through the integration of metagenomic and metabolomic approaches. This would enable the identification of alterations in functional genes and metabolic products associated with *Aiptasia*'s microbial community. Such an approach could provide direct evidence for uncovering the adaptive mechanisms of *Aiptasia*'s symbiotic functional body in response to antibiotic stress.

We agree with the reviewer that integration of metagenomic and metabolomic approaches will help to advance our understanding of the impact of antibiotics on microbial function and

consequently host performance. Our study set out to first quantify the impacts of antibiotics on microbial load and host fitness over time. Our primary goal was to show that we could generate individuals with lower microbial load and, ultimately, reduced diversity. Future experiments will leverage other 'omics approaches to define the role of individual microbial species on symbiotic function. Please see our responses to Reviewer #2 points 4 and 5 below.

Materials and Methods

1. Lines 147-149: The selection of ABS1 (carbenicillin, chloramphenicol, nalidixic acid, and rifampicin) and ABS2 (neomycin, penicillin, rifampicin, streptomycin) as antibiotics for treatment needs clarification. What do these antibiotics represent, and can their concentrations in natural water reach the levels used in the experiment? These considerations are pivotal for assessing the accuracy of the experimental methods.

We agree with the reviewer that explaining the selection of the antibiotic solutions is important towards determining the quality of the experimental results. We originally had this explanation in the methods section (original at line 614), however we decided to move the statement up in the manuscript to be included in the results section so the reader would be able to better interpret the results. While we would not expect these antibiotics to reach these concentrations in natural seawater, this is not relevant for our study as the major goal was to generate in lab gnotobiotic individuals that can be used to better define microbial interactions within the anemone host rather than explore natural patterns in cnidarian microbiomes.

We have moved the following statement to Line 153:

“We chose to use two different solutions in order to (1) determine if one solution is more effective than the other, (2) compare differences in microbial community composition as the different types of antibiotics have varying mechanisms of action to remove microbial species, and (3) because both solutions are known to deplete cnidarian microbiomes (31, 50).”

2. Lines 146-150: The durations of artificial seawater treatment, antibiotic exposure, and the recovery phase were 33 days, 22 days, and 21 days, respectively. What is the rationale behind choosing different durations for these experimental phases?

We agree with the reviewer it's important to provide information regarding the rationale of the timing of our experimental phases. We have added line 157 to the manuscript which states:

“The timing of the experimental phases was based on prior work in anemones showing that priming in FASW for 33-days followed by a 22-day antibiotic exposure is successful in knocking down the Aiptasia microbiome (Dugan et al. 2021, Hartmann et al. 2022, Costa et al. 2021). We chose to follow the anemones for three weeks during recovery to determine the point at which the microbiome recovered to control levels.”

Discussion

3. Lines 528-539: Exploring whether maintaining low microbial diversity in *Aiptasia* over the long term is beneficial or detrimental to the health of the symbiotic functional body would benefit from a more in-depth discussion, possibly integrating an open microbial system with the "Anna Karenina" principle.

We thank the reviewer for pointing out the potential for our work to fit within the ideas proposed by the Anna Karenina hypothesis for host microbiomes. We have added the following lines 565-577 to the discussion to place our work into context of this theory:

*"One benefit to maintaining a stable but low diversity microbiome is that we can begin to explore the potential for destabilization of the function of the symbiosis. In our study, we observed that less diverse individuals had reduced fitness in comparison to the controls. However, fitness metrics only started to rebound once the microbiome stabilized in the third week of microbiome recovery. These data support predictions described within the Anna Karenina Principle for animal microbiomes. The Anna Karenina Principle predicts that disturbance events, such as antibiotic exposure, will have a stochastic effect on the animal microbiome leading to microbiome dispersion that will negatively impact host function (Zaneveld et al. 2017 Nat Micro). Thus, microbiome dispersion, rather than alpha diversity alone, plays a key role in defining host health and metabolism (Zaneveld et al. 2017 Nat Micro). While it was beyond the scope of our study to explicitly test the Anna Karenina hypothesis, future experiments building from our work can leverage gnotobiotic *Aiptasia* to better explore dysbiosis of cnidarian systems."*

1. Zaneveld, J., McMinds, R. & Vega Thurber, R. Stress and stability: applying the Anna Karenina principle to animal microbiomes. *Nat Microbiol* **2**, 17121 (2017).
<https://doi.org/10.1038/nmicrobiol.2017.121>

4. The author leaves several questions unanswered in the discussion, such as "However, it is unclear how the reduction of the microbiome will impact metaorganism response to subsequent disturbance events" (lines 588-589) and "The role and metabolisms of these ASVs within *Aiptasia* are currently unknown and remain targets for cultivation and cultivation-free analysis" (lines 559-561). Addressing these questions could be achieved with additional experimental work.

*We agree with the reviewer that many of our observations require additional experiments to further explore. Currently, we are cultivating members of the *Aiptasia* microbiome and implementing metagenomic approaches to better quantify and describe the metabolism of many of these microbial species, however this additional work goes beyond the scope of the current manuscript. Indeed, our paper is centered on determining the impact of microbiome removal on *Aiptasia* fitness to first quantify the impacts of the total community on host health*

and to determine if it is possible to develop a gnotobiotic model where we can begin to manipulate community composition and test these unanswered questions.

In response to both reviewer concerns, we have made the following modifications:

Line 597:

“The role and metabolisms of these ASVs within Aiptasia are currently unknown and it is unclear if they are beneficial to the host or cause disease. Consequently, they remain targets for cultivation and cultivation-free analysis (e.g., metagenomics and metatranscriptomics) to determine their putative role within the Aiptasia host.”

Line 630:

“Further experiments leveraging multiple exposures to disturbance are needed to quantify the protective role of the host microbiome to stress.”

5. The findings appear to echo patterns observed in previous research on cnidarians. Utilizing a multi-omics approach for the model organism Aiptasia could provide a more comprehensive understanding of the response and adaptation patterns of the symbiotic functional body to antibiotic treatment.

We agree with the reviewer that our work echo some patterns in other cnidarian species, however we also showcase where we have extended past work. Namely, we show that it is indeed feasible to generate a gnotobiotic model using Aiptasia, which is one of the most widely used cnidarians for studying coral symbioses. Our work extends past research on the application of antibiotics in cnidarians by highlighting that microbiome knockdown and recovery is a dynamic process that requires a longitudinal approach to realize the impacts on both the microbial community and host fitness. Because our goal was to first explore microbiome dynamics as they relate to host fitness, we chose to not collect multi-omics data in this context. We agree with the reviewer that a multi-omics approach is required to begin to define some of the mechanisms underlying the response and adaption of Aiptasia in response to antibiotic treatment (and subsequent microbial re-introduction studies), but this was beyond the scope of this paper.

To reflect these ideas, we have added line 631 to our discussion:

“Moreover, it is essential to combine these studies with multi-omics approaches to determine mechanisms leading to microbiome disturbance and recovery following antibiotic exposure.”

Re: mSystems01342-23R1 (**Microbiome depletion and recovery in the sea anemone, *Exaiptasia diaphana*, following antibiotic exposure**)

Dear Dr. E. Maggie Sogin:

Your manuscript has been accepted, and I am forwarding it to the ASM production staff for publication. Your paper will first be checked to make sure all elements meet the technical requirements. ASM staff will contact you if anything needs to be revised before copyediting and production can begin. Otherwise, you will be notified when your proofs are ready to be viewed.

Cover Image Submissions: If you would like to submit a potential Cover Image, please email a file and a short legend to msystems@asmusa.org. Please note that we can only consider images that (i) the authors created or own and (ii) have not been previously published. By submitting, you agree that the image can be used under the same terms as the published article. Image File requirements: TIF/EPS, 7.5 inches wide by 8.25 inches tall (at least 2,250 pixels wide by 2,475 pixels tall), minimum 300 dpi resolution (600 dpi preferred), RGB, and no figure elements, e.g., arrows or panel labels. The legend should be a short description of the image, 1-2 sentences recommended.

Sincerely,
Laetitia Wilkins

Editor
mSystems

Reviewer #2 (Comments for the Author):

The authors have diligently refined the manuscript based on the feedback received. As a result, the overall quality of the manuscript has been significantly enhanced. This demonstrates their commitment to producing a high-caliber piece of work and ensuring that it meets the necessary standards.